# Secondary organic aerosol formation from in situ OH, O$_3$, and NO$_3$ oxidation of ambient forest air in an oxidation flow reactor

Brett B. Palm[1,2], Pedro Campuzano-Jost[1,2], Douglas A. Day[1,2], Amber M. Ortega[1,3,+], Juliane L. Fry[4], Steven S. Brown[2,5], Kyle J. Zarzana[1,2,*], William Dube[1,5], Nicholas L. Wagner[1,5], Danielle C. Draper[4,#], Lisa Kaser[6], Werner Jud[7,^], Thomas Karl[8], Armin Hansel[7], Cándido Gutiérrez-Montes[9], and Jose L. Jimenez[1,2]

[1]Cooperative Institute for Research in Environmental Sciences, University of Colorado, USA
[2]Department of Chemistry and Biochemistry, University of Colorado, USA
[3]Department of Atmospheric and Oceanic Science, University of Colorado, USA
[4]Department of Chemistry, Reed College, USA
[5]NOAA Earth System Research Laboratory, Chemical Sciences Division, Boulder, CO, USA
[6]Atmospheric Chemistry Observations & Modeling Laboratory, National Center for Atmospheric Research, USA
[7]Institute of Ion Physics and Applied Physics, University of Innsbruck, Austria
[8]Institute of Atmospheric and Cryospheric Sciences, University of Innsbruck, Austria
[9]Departamento de Ingeniería, Mecánica y Minera, Universidad de Jaen, Jaen, Spain

[+]Now at Air Pollution Control Division, Colorado Department of Public Health and Environment, Denver, CO, USA
[*]Now at NOAA Earth System Research Laboratory, Chemical Sciences Division, Boulder, CO, USA
[#]Now at Department of Chemistry, University of California, Irvine, USA
[^]Now at Research Unit Environmental Simulation (EUS), Institute of Biochemical Plant Pathology (BIOP), Helmholtz Zentrum München GmbH, Germany

*Correspondence to:* Jose L. Jimenez (jose.jimenez@colorado.edu)

**Abstract:** Ambient pine forest air was oxidized by OH, O$_3$, or NO$_3$ radicals using an oxidation flow reactor (OFR) during the BEACHON-RoMBAS (Bio-hydro-atmosphere interactions of Energy, Aerosols, Carbon, H$_2$O, Organics & Nitrogen–Rocky Mountain Biogenic Aerosol Study) campaign to study biogenic secondary organic aerosol (SOA) formation and organic aerosol (OA) aging. A wide range of equivalent atmospheric photochemical ages was sampled, from hours up to days (for O$_3$ and NO$_3$) or weeks (for OH). Ambient air processed by the OFR was typically sampled every 20-30 min, in order to determine how the availability of SOA precursor gases in ambient air changed with diurnal and synoptic conditions, for each of the three oxidants. More SOA was formed during nighttime than daytime for all three oxidants, indicating that SOA precursor concentrations were higher at night. At all times of day, OH oxidation led to approximately 4 times more SOA formation than either O$_3$ or NO$_3$ oxidation. This is likely because O$_3$ and NO$_3$ will only react with gases containing C=C bonds (e.g., terpenes) to form SOA, but won't react appreciably with many of their oxidation products or any species in the gas phase that lacks a C=C bond (e.g., pinonic acid, alkanes). In contrast, OH can continue to react with compounds that lack C=C bonds to produce SOA. Closure was achieved between the amount of SOA formed from O$_3$ and NO$_3$ oxidation in the OFR and the SOA predicted to form from measured concentrations of ambient monoterpenes and sesquiterpenes using published chamber yields. This is in contrast to previous work at this site (Palm et al., 2016), which has shown that a source of SOA from semi- and intermediate-volatility organic compounds (S/IVOCs) 3.4 times larger than the source from measured VOCs is needed to explain the measured SOA formation from OH oxidation. This work suggests that those S/IVOCs typically do not contain C=C bonds. O$_3$ and NO$_3$ oxidation produced SOA with elemental O:C and H:C similar to the least oxidized OA observed in local ambient air, and neither oxidant led to net mass loss at the highest exposures, in contrast with OH oxidation. An OH exposure in the OFR equivalent to several hours of atmospheric aging also produced SOA with O:C and H:C values similar to ambient OA, while higher aging (days–weeks) led to formation of SOA with progressively higher O:C and lower H:C (and net mass loss at the highest exposures). NO$_3$ oxidation led to the production of particulate organic nitrates (pRONO$_2$), while OH and O$_3$ oxidation (under low NO) did not, as expected. These measurements of SOA formation provide the first direct comparison of SOA formation potential and chemical evolution from OH, O$_3$ and NO$_3$ oxidation in the real atmosphere, and help to clarify the oxidation processes that lead to SOA formation from biogenic hydrocarbons.

# 1    Introduction

Submicron atmospheric aerosols have important impacts on radiative climate forcing (Myhre et al., 2013) and human health (Pope and Dockery, 2006). A large fraction of submicron particulate mass is composed of organic aerosols (OA), and is produced from a variety of sources (Zhang et al., 2007). Primary OA (POA) is directly emitted as particles (e.g., via fossil fuel combustion, biomass burning), while secondary OA (SOA) can be formed through gas-phase oxidation and gas-to-particle conversion of directly emitted organic gases, or via aqueous pathways. Globally, SOA comprises the majority of OA, particularly in rural locations away from primary sources (Zhang et al., 2007; Jimenez et al., 2009). However, the processes of formation, chemical transformation, and removal of SOA remain uncertain (Hallquist et al., 2009; Shrivastava et al., 2016).

Hydroxyl radicals (OH), ozone ($O_3$), and nitrate radicals ($NO_3$) are the three major oxidants in the atmosphere that react with organic gases to form SOA. The initial steps of oxidation for each oxidant are summarized here according to Atkinson and Arey (2003):

- OH can react via H-abstraction or addition to a C=C double bond, depending on the structure of the organic molecule;
- $O_3$ generally reacts only with alkenes, adding to a C=C bond to produce a primary ozonide which then decomposes to form a carbonyl plus a Criegee intermediate;
- $NO_3$ radicals also react by addition to a C=C bond, producing an organic peroxy radical with an adjacent organic nitrate group that will react further. The nitrate functional group formed during the initial $NO_3$ addition can either remain in the product molecule or decompose to produce $NO_2$ (g).

Nearly all oxidation pathways in the atmosphere will lead to the production of a peroxy radical ($RO_2$), which can proceed to react with $HO_2$, $NO_2$, NO, another $RO_2$, or undergo autooxidation (Atkinson, 1997; Orlando and Tyndall, 2012; Crounse et al., 2013). Reaction rate constants and more detailed reaction mechanisms can be found elsewhere (e.g., Atkinson et al., 1982; Atkinson, 1997; Chew et al., 1998; Calvert et al., 2002).

SOA yields from the oxidation of a wide variety of precursor gases by each of these three oxidants have been reported. SOA yields are typically measured from oxidation experiments in large environmental chambers. These yields are evaluated through implementation in regional or global models, which can be compared to

ambient measurements (e.g., Volkamer et al., 2006; Hayes et al., 2015). However, large chamber experiments have been shown to be affected by large losses of semivolatile and low volatility gases (Matsunaga and Ziemann, 2010; Zhang et al., 2014; Krechmer et al., 2015; La et al., 2016; Nah et al., 2016) and particles (Crump and Seinfeld, 1981; McMurry and Rader, 1985; Pierce et al., 2008) to the chamber walls. These artifacts affect

the ability to accurately measure SOA yields, and also limit the amount of oxidation that can be achieved in chambers. Large variability in OA concentrations exists between various global OA models, which typically achieve poor agreement and correlation with ambient surface and vertical profile OA concentration measurements (Tsigaridis et al., 2014).

In addition to bulk concentrations, the chemical composition of OA also determines its atmospheric properties.

The elemental O:C and H:C ratios of OA can be measured using aerosol mass spectrometry (Aiken et al., 2008; Canagaratna et al., 2015). The O:C and H:C ratios can provide information about the sources and evolution of OA in the atmosphere (Aiken et al., 2008; Heald et al., 2010; Kroll et al., 2011; Ng et al., 2011), and also often correlate with key OA properties such as hygroscopicity, material density, and phase separation (Jimenez et al., 2009; Bertram et al., 2011; Kuwata et al., 2012). Laboratory studies have typically struggled to reproduce the

O:C and H:C values found in ambient OA, particularly for the highest O:C values found in remote areas (Aiken et al., 2008; Chen et al., 2015).

While large chambers have been the standard method for studying SOA yields and composition, and are the basis for parameterized yields and oxidation in most models, oxidation flow reactors (OFRs) have recently become a popular alternative approach. OFRs typically have shorter residence times than chambers, which

reduces wall contact. Also, ambient air can easily be oxidized in an OFR, while it is difficult and slow to perform such experiments in a large chamber (Tanaka et al., 2003). SOA yields from OH oxidation in OFRs for a variety of individual and mixed precursors have been reported, and generally show that yields in OFRs are similar to chamber yields (Kang et al., 2007, 2011, Lambe et al., 2011, 2015; Li et al., 2013; Bruns et al., 2015). Properties related to SOA elemental composition have also been investigated in OFRs (Massoli et al., 2010; Lambe et al.,

2011, 2012, 2014; Saukko et al., 2012; Ortega et al., 2013, 2016). However, these studies were limited to laboratory-produced SOA from one or several precursor gases, often at very high concentrations. Several studies have reported on SOA formation from the OH oxidation of ambient air (Ortega et al., 2016; Palm et al., 2016) or emission sources (Cubison et al., 2011; Keller and Burtscher, 2012; Ortega et al., 2013; Tkacik et al.,

2014; Bruns et al., 2015; Karjalainen et al., 2016; Timonen et al., 2016), but SOA from $O_3$ and $NO_3$ oxidation of ambient air or direct source emissions has not been studied using an OFR, to our knowledge.

In this study, we oxidized ambient pine forest air with either OH, $O_3$, or $NO_3$ in an OFR to investigate how much SOA can be formed from real ambient mixtures of largely biogenic SOA precursor gases, how the SOA precursor concentrations varied with time, and the properties of the SOA formed. The amount of SOA formed from each oxidant was compared to the amount predicted to form from oxidation of the measured ambient VOCs that entered the OFR. We investigated the elemental composition of the SOA that was formed as a function of the amount of oxidant exposure (oxidant concentration multiplied by residence time) in the OFR. The contribution of organic nitrate to SOA formation was also explored and compared to the results with ambient and chamber studies.

## 2      Experimental methods

### 2.1      BEACHON-RoMBAS field campaign

The OFR measurements presented here were conducted during July–August 2011 as part of the BEACHON-RoMBAS field campaign (Bio-hydro-atmosphere interactions of Energy, Aerosols, Carbon, $H_2O$, Organics & Nitrogen – Rocky Mountain Biogenic Aerosol Study; http://cires.colorado.edu/jimenez-group/wiki/index.php/BEACHON-RoMBAS). The research site was located in a ponderosa pine forest in a mountain valley at the Manitou Experimental Forest Observatory, near Woodland Park, Colorado (39.10° N, 105.10° W; 2370 m elevation). An overview of previous research at this site, including BEACHON-RoMBAS and prior campaigns, has been presented in detail by Ortega et al. (2014). Here we present a brief summary of research site details that are relevant to this analysis.

VOC concentrations at the site (quantified using proton-transfer-reaction time-of-flight mass spectrometry; PTR-TOF-MS) varied on a diurnal cycle, dominated by 2-methyl-3-buten-2-ol (MBO) during daytime and monoterpenes (MT) during nighttime. Fry et al. (2013) and Palm et al. (2016) show diurnal cycles of select biogenic and anthropogenic VOCs. VOC measurements from a July–September 2008 campaign at the same site have also been described in Kim et al. (2010). During BEACHON-RoMBAS, the concentration of MBO+isoprene in the forest canopy ranged from about 2 ppb during daytime to 0.4 ppb at nighttime (see Palm

et al., 2016). The ratio of isoprene to MBO at this pine forest site was determined using $NO^+$ reagent ion chemical ionization mass spectrometry (Karl et al., 2012) and using GC-MS (Kaser et al., 2013) to be about 21%, indicating the concentration of isoprene at this site was low (<0.3 ppb). MT concentrations in the canopy spanned from 0.4 ppb during the day to 1.1 ppb at night, on average. The Manitou Experimental Forest

Observatory site is mainly influenced by biogenic emissions, but occasionally receives airflow from nearby urban areas (Denver metropolitan area and Colorado Springs, 75 and 35 km away from the site respectively), leading to moderate increases in $NO_x$ (up to ~5 ppbv from ~2 ppbv), CO (up to ~140 ppbv from ~100 ppbv), and anthropogenic VOCs (e.g., benzene up to ~50 pptv from ~20 pptv, and toluene up to ~150 pptv from ~50 pptv) during the late afternoon and evening (Fry et al., 2013; Ortega et al., 2014).

**2.2    OFR methods**

The OFR used in this study was the Potential Aerosol Mass (PAM) flow reactor (Kang et al., 2007, 2011). The PAM reactor is a cylindrical tube 45.7 cm long and 19.7 cm ID with a volume of approximately 13 liters. This type of OFR has been used to study SOA formation and chemistry in a number of previous studies (e.g., Kang et al., 2007, 2011; Massoli et al., 2010; Lambe et al., 2012, 2015; Li et al., 2013; Ortega et al., 2013, 2016; Tkacik et al.,

2014; Palm et al., 2016). During BEACHON-RoMBAS, ambient air was sampled through a 14 cm diameter opening on one end of the OFR (with the inlet plate removed to prevent loss of gases/particles on inlet surfaces) through a coarse-grid mesh screen coated with an inert silicon coating (Sulfinert by Silcotek, Bellefonte, PA). The OFR was located on top of the measurement trailer in order to sample ambient air directly without using an inlet. Therefore the temperature and RH inside the OFR were the same as ambient conditions, with the

exception of minor heating from the UV lamps mounted inside the OH-OFR (up to ~2°C heating at the highest lamp settings, and ~0.5°C at the settings producing the most SOA; Li et al., 2015). No heating occurred during $O_3$ or $NO_3$ modes. Thus RH within the OFR was the same or slightly lower than ambient, depending on the operating mode. The OFR was operated with a residence time in the range of 2–4 min. The residence time distribution in the OFR, modeled using FLUENT for the configuration used in this study (inlet plate removed), is

shown in Fig. S1. The modeled residence time distribution is much more homogeneous than has been measured for OFRs operated with an inlet plate (Lambe et al., 2011; Ortega et al., 2016). However, local winds can result in some variations that are not captured by the FLUENT model. The data were not screened for high local wind speeds. However, periods of high wind speeds were infrequent during the campaign, and the influence of local

winds was likely tempered by the fact that the OFR was located within the canopy of the forest. Two OFRs were used simultaneously, with one dedicated to $NO_3$ oxidation while the other was used for either OH or $O_3$ oxidation. OH radicals were produced in situ inside the OFR using two different methods, referred to as OFR185 and OFR254 (named according to the wavelength of the highest energy UV light used to generate oxidants within the reactor). These methods have been described in detail previously and showed consistent results (Palm et al., 2016). All results of OH oxidation presented in this paper used the OFR185 method. The gas-phase $HO_x/O_x$ chemistry and possible non-OH chemistry inside the OFR was investigated with kinetic modeling (Li et al., 2015; Peng et al., 2015, 2016). For the wide variety of compounds investigated in Peng et al. (2016), reactions with OH dominated over other possible reactions, including $O(^1D)$, $O(^3P)$, $O_3$, and photolysis at 185 nm or 254 nm, under the conditions of OH oxidation in the OFR during this campaign.

$NO_3$ radicals were generated by thermal decomposition of $N_2O_5$ ($N_2O_5 \rightarrow NO_2 + NO_3$), which was injected into the OFR from a cold trap held in a dry ice + isopropyl alcohol bath. The cold trap was held near -60°C using a temperature controlled copper sleeve immersed in the -78° C bath. A 10–100 sccm flow of zero air eluted $N_2O_5$ from the trap. This $N_2O_5$+zero air mixture was injected through an approximately 14 cm diameter ring of 1/8" Teflon tubing with pinholes around the ring mounted just inside the OFR entrance inside the mesh screen. $N_2O_5$ concentrations were adjusted by changing this flow rate from the $N_2O_5$ dry ice reservoir. The concentrations of $N_2O_5$ and $NO_3$ in both the injection flow and in the output of the OFR were measured using diode laser-based cavity ring-down spectroscopy (CRDS; Wagner et al., 2011). The concentration of $NO_2$ was measured in the output of the OFR using laser-induced fluorescence (Thornton et al., 2000). The experimental setup for the $NO_3$-OFR system is illustrated in Fig. S2 and discussed in Sect. S1.

To estimate $NO_3$ concentrations and exposure in the OFR, the relevant chemistry was modeled using a chemical-kinetic plug-flow model, implemented in the KinSim chemical-kinetic integrator (version 3.10) using Igor Pro 6 (http://www.igorexchange.com/node/1333; Wavemetrics, Lake Oswego, OR, USA). A key output of this model was the integrated $NO_3$ exposure experienced by MT-containing air during the OFR residence time, calculated as the integral of $NO_3$ concentration over the OFR residence time (in units of molecules $cm^{-3}$ s), and multiplied by the fraction of MT that was estimated to have been mixed with the $N_2O_5$ flow at each residence time, due to lack of mixing from the small flow rate (see Sect. S1 for more details of the unmixed fraction estimation and parameterization). $NO_3$ exposure was converted to an equivalent (eq.) atmospheric age by

dividing by a typical site-specific nighttime ambient $NO_3$ concentration, which has been estimated to be on the order of 1 ppt (Fry et al., 2013). This eq. age represents the amount of time the air would have to spend in the atmosphere with 1 ppt $NO_3$ to experience the same amount of $NO_3$ exposure as in the OFR. The unit of eq. age is a unit of exposure. When given in units of eq. days, it represents the number of 24 h periods that air would

need to spend in an atmosphere containing the stated oxidant concentration in order to achieve the equivalent amount of exposure as in the OFR (which applies for OH and $O_3$ eq. ages as well). More details about the model can be found in Sect. S1.

The exposure metric for the $NO_3$-OFR is specific to the site in which it is measured. Fry et al. (2013) estimated the average nighttime $NO_3$ concentration at this site (approximately 1 pptv) from an average $NO_3$ production

rate and lifetime of approximately 0.03 pptv $s^{-1}$ and 25 s, respectively. Other sites can have considerably different production rates for $NO_3$ and thus very different nightime exposures. Remote forests, with nighttime $NO_x$ below 50 pptv, could experience $NO_3$ production rates more than 10 times slower, while forests immediately downwind of urban areas could have $NO_3$ production rates more than 10 times faster (e.g., outflow from Houston, TX; Brown et al., 2013). Variability in $NO_3$ production rates and observed $NO_3$ levels is a common

feature of recent field observations (Brown and Stutz, 2012). Estimated eq. $NO_3$ ages from this study are therefore shown simply for a common metric of comparison for all of the data during this study, interpretable in terms of the average chemistry occurring at the BEACHON site. Interpretation of measurements at other sites would need to be adjusted to local $NO_3$ concentrations.

To investigate SOA formation from $O_3$ oxidation, $O_3$ was produced external to the OFR by flowing pure dry $O_2$

gas across two low-pressure mercury UV lamps (BHK, Inc., model no. 82-9304-03). The $O_2$ was photolyzed by 185 nm light to produce $O(^3P)$, which further reacted with $O_2$ to produce $O_3$. This $O_2+O_3$ mixture was injected at 0.5 lpm into the front of the OFR through four ports distributed evenly around and just inside the 14 cm opening. $O_3$ concentrations were cycled by adjusting the UV lamp intensity (i.e., photon flux) in the $O_3$ generation setup. $O_3$ was measured in the output of the OFR using a 2B Technologies Model 205 Monitor. $O_3$

exposure was calculated by multiplying the measured $O_3$ concentration in the OFR output by the residence time of the OFR. Loss of injected $O_3$ to internal OFR walls was not investigated, so the exposure may be slightly underestimated by this method. $O_3$ exposure was converted to an eq. atmospheric age by dividing by a typical,

site-specific, 24 h average, ambient $O_3$ concentration of 50 ppb. A schematic of the $O_3$-OFR system is also shown in Fig. S2.

## 2.3 Particle and gas measurements

Ambient and OFR-oxidized particles were measured with an Aerodyne High-Resolution Time-of-Flight Aerosol
Mass Spectrometer (HR-ToF-AMS, referred to here as AMS; DeCarlo et al., 2006; Canagaratna et al., 2007) and a TSI 3936 Scanning Mobility Particle Sizer (SMPS). Details of these measurements have been described previously (Palm et al., 2016). Ambient VOC concentrations were quantified using a PTR-TOF-MS (Kaser et al., 2013). The OFR output was sampled by the PTR-TOF-MS during selected periods only (Aug 4–6, 9–10, and 22–23 for $NO_3$ oxidation, and Aug. 7–9 and 23–24 for $O_3$ oxidation; see Palm et al. (2016) for details of sampling VOCs
during OH oxidation). The particle mass measurements were corrected for particle losses to sampling line walls and at the small particle transmission limit of the AMS aerodynamic lens (combined 2% correction; details of these corrections are the same as in Palm et al., 2016). To account for particle losses to internal OFR surfaces, the particle mass was corrected by the average ratio of ambient particle mass to the particle mass measured through each OFR in the absence of oxidant (1% correction for the $O_3$ OFR, and 14% for the $NO_3$ OFR due to a
different sampling port with a higher wall surface-area-to-volume ratio).

A correction was also applied to account for any condensable oxidation products (referred to as low-volatility organic compounds; LVOCs) that were formed from gas-phase oxidation in the OFR but condensed on OFR or sampling line walls instead of condensing to form SOA. This is non-atmospheric behavior, due to the short residence time in the OFR and the relatively small aerosol condensational sink in this study. A correction is
needed because the dominant fate of such gases in the atmosphere will be condensation to form SOA (lifetime of ~minutes) rather than being lost to any environmental surfaces via dry or wet deposition (lifetime of ~hours to a day; Farmer and Cohen, 2008; Knote et al., 2015; Nguyen et al., 2015). This correction, referred to as the "LVOC fate correction", was first represented in a model developed in Palm et al. (2016); the full details of the model can be found there. Briefly, the model takes several inputs, including particle condensational sink, OFR
residence time, and oxidant concentration. It produces the fractional fates of LVOCs with respect to condensation onto particles, condensation onto OFR walls, further oxidation to give non-condensable molecular fragmentation products, and condensation onto sampling line walls after exiting the back of the OFR. In Palm et

al. (2016), the model was verified by quantitatively explaining $SO_4$ aerosol formation from OH oxidation of ambient $SO_2$.

The results of the LVOC fate model for the $O_3$-PAM and $NO_3$-PAM conditions in this study are shown in Fig. S3. The SOA formation values given in the subsequent analysis are corrected for LVOC fate by dividing the

measured SOA formation by the fraction of LVOCs predicted to have condensed to form SOA in the OFR (an average correction of 0.4 $\mu g\ m^{-3}$ for both $O_3$-PAM and $NO_3$-PAM). These corrected values refer to the amount of SOA that would form from any ambient precursors in the absence of OFR walls and the limited time for condensation within the OFR. LVOCs are assumed not to be lost to fragmentation from excessive $O_3$ or $NO_3$ reactions in the gas-phase prior to condensation due to lack of C=C bonds (which is different from the

parameterization for OH reactions used in Palm et al., 2016). This assumption is reinforced by the fact that for the highest $O_3$ and $NO_3$ eq. ages achieved in this work, no net decrease of OA was observed when SOA-forming gases were not present (see Sect. 3.2.1 and Fig. 5). If fragmentation reactions in the gas phase (or from heterogeneous oxidation) were important for the range of eq. ages studied here, observations would show a net loss of OA at the highest eq. ages when SOA-forming gases (e.g., MT) were not present.

**2.4    Modeling of SOA formation**

In the analysis in Sect. 3.2.2, the amount of SOA formed by oxidation of ambient air by $O_3$ or $NO_3$ in the OFR is compared to the amount predicted to form. This predicted amount was estimated by applying SOA yields to the fraction of measured ambient MT and sesquiterpenes (SQT) concentrations that were predicted to react. Since the ambient VOC measurements were taken above the canopy at a height of 25 m, the concentrations were

corrected to reflect in-canopy values that were ingested into the OFR, a technique which has been used previously (Kim et al., 2013; Wolfe et al., 2014; Palm et al., 2016). During this campaign, speciated MT and SQT measurements were not available. When predicting SOA formation in this analysis, we use previous measurements at the same site to approximate that MT consisted of an equal mix of $\alpha$-pinene, $\beta$-pinene, and 3-carene and that SQT was solely isolongifolene (Kim et al., 2010). Numerous chamber studies have reported SOA

yields of individual MT from $O_3$ oxidation (e.g., Ng et al., 2006; Pathak et al., 2007, 2008; Shilling et al., 2008; Zhao et al., 2015) and from $NO_3$ oxidation (Hallquist et al., 1999; Moldanova and Ljungström, 2000; Spittler et al., 2006; Fry et al., 2009, 2011, 2014; Boyd et al., 2015; Ng et al., 2016). SOA yields from SQT have also been

reported for $O_3$ oxidation (Jaoui et al., 2003, 2013; Ng et al., 2006; Winterhalter et al., 2009; Chen et al., 2012; Tasoglou and Pandis, 2015) and $NO_3$ oxidation (Fry et al., 2014). In this analysis, the OA concentrations measured after $O_3$ or $NO_3$ oxidation ranged from 1–3 µg m$^{-3}$, with few exceptions. For simplicity with this relatively narrow range, the dependence of SOA yields on OA concentrations was not included. Instead, we

applied representative SOA yields of 15% for ozonolysis of $\alpha$-pinene, $\beta$-pinene, and 3-carene, and 30% for ozonolysis of isolongifolene. For reaction with $NO_3$, SOA yields of 4%, 33%, 38%, and 86% were used for $\alpha$-pinene, $\beta$-pinene, 3-carene, and isolongifolene (using $\beta$-caryophyllene as a proxy for all SQT; Fry et al., 2014; Kang et al., 2016; Ng et al., 2016). The rate constants used for reaction of $\alpha$-pinene, $\beta$-pinene, 3-carene, and isolongifolene with $O_3$ and $NO_3$ were $k_{O_3}$ = $8.6 \times 10^{-17}$, $1.5 \times 10^{-17}$, $3.6 \times 10^{-17}$, and $1.1 \times 10^{-17}$ cm$^3$ molec$^{-1}$ s$^{-1}$,

and $k_{NO_3}$ = $6.1 \times 10^{-12}$, $2.5 \times 10^{-12}$, $9.5 \times 10^{-12}$, and $3.9 \times 10^{-12}$ cm$^3$ molec$^{-1}$ s$^{-1}$, respectively (Canosa-Mas et al., 1999; Atkinson and Arey, 2003; Richters et al., 2015).

## 3      Results and discussion

### 3.1      Modeled vs. measured $NO_3$ and $O_3$ exposures

One of the features of the OFR technique is the short residence time required for conducting high time

resolution ambient measurements. Combined with the ability to rapidly change the amount of oxidant injected or produced in the OFR, this allows for a wide range of oxidation levels to be studied in a short amount of time (and thus with limited variation of ambient conditions). In this work, the oxidant concentration was changed every 20–30 min, covering a range from no added oxidant to maximum oxidation repeatedly in 2–3 h cycles. In order to interpret the results over the wide range of oxidant exposure, the amount of exposure must be

quantified. In Palm et al. (2016), OH exposure was estimated using a model-derived equation (Li et al., 2015; Peng et al., 2015) and calibrated using PTR-TOF-MS measurements of VOC decay in the OFR. In this work, a simple box model was developed and compared with VOC decay measurements to estimate $NO_3$ and $O_3$ exposures in the OFR.

The set of reactions and rate constant parameters included in the modeling of $NO_3$ exposure are shown in Table

S1. Figure 1 illustrates the most important mixing ratios and reactive fluxes in the OFR with injected $N_2O_5$ under typical conditions. Interconversion between $N_2O_5$ and $NO_2$ + $NO_3$ was relatively rapid, which maintained the

system near equilibrium at all times. Wall loss of $N_2O_5$ was estimated to be the main loss of the injected nitrogen-containing species (84%), while reaction of $NO_3$ with biogenic gases (2%), $NO_3$ wall losses (14%), and hydrolysis of $N_2O_5$ on particle surfaces (0.2%) were minor loss pathways. Figure 2a–c compares the $N_2O_5$, $NO_2$, and $NO_3$ mixing ratios measured in the OFR output with those predicted by the model. The model is generally

consistent with the measurements. The scatter in the measurements is thought to be due mainly to incomplete and/or variable mixing of the injected $N_2O_5$ flow into the sampled ambient air (see Sect. S1 for more details), with some contribution from measurement variability at low ambient MT concentrations. The critical output of this model for our application is the prediction of the fraction of MT reacted. Figure 2d shows that the model can reproduce the measured MT decay with an error (average absolute value of modeled minus measured

fraction MT remaining) of 11%, providing confirmation that using the model output $NO_3$ exposure in the subsequent analysis of aerosol mass yields from the OFR is justified. A similar analysis of SQT decay was not possible, because ambient SQT concentrations were too small to accurately measure fractional decays. Also, MBO did not react substantially with $NO_3$ in the OFR, consistent with the lifetime for reaction of $NO_3$ with MBO that is approximately 3 orders of magnitude slower than for reaction with MT (Atkinson and Arey, 2003). This is

also representative of the atmosphere, where MBO will overwhelmingly react with OH or $O_3$ and not $NO_3$ (Atkinson and Arey, 2003).

Unlike $NO_3$ exposure, the estimation of $O_3$ exposure did not require a detailed chemical model since the $O_3$ system had no reservoir species analogous to $N_2O_5$. $O_3$ exposure was simply estimated as the measured $O_3$ concentration in the OFR output multiplied by residence time. To verify this estimate, the measured fraction of

MT that reacted in the OFR was compared in Fig. 3 to a model prediction calculated using a simple set of reactions of ozone with the three major MT species (Table S2). The model is consistent with measurements within an error of 9%, and shows that a parameterization for mixing of the $O_3$ flow into ambient air was not needed. In contrast to the slower 10–100 sccm flow of $N_2O_5$, the 0.5 lpm flow of $O_2+O_3$ appears to have been large enough relative to the total OFR flowrate to result in sufficiently complete mixing. This result suggests that

a faster flow of $N_2O_5$ could be used in future $NO_3$ oxidation experiments to facilitate better mixing.

Time series examples of measured and modeled MT remaining after OFR oxidation are compared to ambient MT concentrations for both $NO_3$ and $O_3$ oxidation in Fig. 4. These examples illustrate the dynamic range from no

MT reacted (i.e., when no oxidants were added to the ambient air) to nearly all MT reacted within the 2–3 h cycles for both oxidants. Further examples are shown for $NO_3$ oxidation in Fig. S6 and for $O_3$ oxidation in Fig. S7.

## 3.2    SOA formed from oxidation of ambient air

### 3.2.1    OA enhancement vs. photochemical age

During BEACHON-RoMBAS, ambient air was oxidized by either OH, $O_3$, or $NO_3$ in order to study the amount and properties of SOA that could be formed from ambient precursors. In situ SOA formation from OH oxidation was the subject of a previous manuscript (Palm et al., 2016). Select results are reproduced here as a comparison to SOA formation from $O_3$ and $NO_3$ oxidation. Additional new analyses of the chemical composition of SOA formed from OH oxidation is also included along with $O_3$ and $NO_3$ oxidation in Sects. 3.3-3.4.

In Palm et al. (2016), SOA formation from OH oxidation in the OFR correlated with ambient MT concentrations (and implicitly with any other gases that correlated with MT, such as SQT and possibly terpene oxidation products). Here, Fig. 5 shows the OA enhancement observed after $O_3$ and $NO_3$ oxidation as a function of eq. age in the OFR. Similar to OH oxidation, little SOA formation was observed from $O_3$ or $NO_3$ oxidation when ambient MT concentrations were low, regardless of the amount of exposure. When MT concentrations were higher, increasing amounts of SOA were formed with increasing exposure. As seen in Fig. 5 (and in Fig. 6 below), lower eq. $NO_3$ ages were achieved when MT concentrations were higher, and higher eq. $NO_3$ ages were achieved when MT concentrations were lower. This was because the higher MT concentrations occurred during nighttime, when lower ambient temperatures shifted the equilibrium towards $N_2O_5$ and away from $NO_2+NO_3$ (from the injected $N_2O_5$), meaning lower $NO_3$ exposures were realized in the OFR.

Another way to examine the trends in OA enhancement is by separating the results into daytime and nighttime. Due to diurnal cycles in the emission rates (that are strong functions of temperature, and also light for some species), vertical mixing in the boundary layer, and changing rates of ambient oxidation, the concentration of MT (and other SOA precursors) in ambient air showed substantial diurnal cycles (Kim et al., 2010; Fry et al., 2013; Kaser et al., 2013). Ambient air was characterized by higher MBO+isoprene (with ambient OH and $O_3$ chemistry) during the day and higher MT+SQT (with ambient $O_3$ and $NO_3$ chemistry) during the night (Fry et al., 2013). Due to these changes, it might be expected that SOA formation in the OFR would also change diurnally.

OA enhancements vs. eq. age for OH, $O_3$, and $NO_3$ oxidation are shown together in Fig. 6, split between daytime (08:00-20:00 LT) and nighttime (20:00-08:00 LT). For all oxidants, more SOA formation was observed during nighttime. This is consistent with the general increase in MT and SQT (average of 1.1 and 0.04 ppbv in the canopy during nighttime, and 0.4 and 0.03 ppbv during daytime, respectively) and related precursor concentrations in the shallower nighttime boundary layer. This higher SOA formation during nighttime was not a result of larger temperature-dependent partitioning to the particle phase at lower nighttime temperatures, as evidenced by stable values of measured OA enhancement per unit ambient MT (the dominant measured SOA precursor) across the whole range of ambient temperatures (shown in Fig. S8). An exploration of the correlation between maximum SOA formation from each oxidant and all available ambient VOC concentrations is shown in Fig. S9, illustrating that MT are the best tracer of SOA production at this forest site. The maximum amount of SOA formed from OH oxidation was approximately 4 times more than from $O_3$ or $NO_3$ oxidation for both daytime and nighttime over the eq. ages covered in this work. If the gases that formed SOA from each oxidant were the same, then this would require the SOA yields from OH oxidation to be more than 4 times larger than from $O_3$ or $NO_3$ oxidation. The references for SOA yields from $O_3$ and $NO_3$ oxidation presented herein and for OH oxidation presented in Palm et al. (2016) show this is likely not the case. Instead, one possible explanation for this result could be that a large fraction of SOA-forming gases found in ambient air do not have C=C bonds (e.g., MT oxidation products such as pinonic acid). Such molecules would typically not react appreciably with $O_3$ or $NO_3$ over the range of eq. ages achieved in this work, but will still react with OH and may lead to SOA formation. Future $O_3$ and $NO_3$ oxidation studies should include higher eq. age ranges in order to investigate if additional SOA could be formed from ambient precursors at higher ages. This concept will be discussed further in Sect. 3.2.2.

Whereas a net loss of OA was observed at >10 eq. days of OH aging due to heterogeneous oxidation (shown in Fig. 7 of Palm et al., 2016), a similar net loss of OA at the highest eq. ages of $O_3$ and $NO_3$ oxidation was not observed. Since the highest eq. ages for both $O_3$ and $NO_3$ oxidation were approximately 5 days, it is unclear if $O_3$ or $NO_3$ heterogeneous oxidation would lead to net loss of ambient OA at substantially higher ages. Future experiments could be designed to achieve higher ages in order to investigate this effect.

### 3.2.2    Measured vs. predicted OA enhancement

When ambient air is sampled into an OFR, any gases or particles present in that air are subject to oxidation. Measurement of the resultant SOA formation is a top-down measure of the total SOA formation potential of that air as a function of eq. age of oxidation. In other words, an OFR can be used to determine the relative concentrations of SOA-forming gases present in ambient air at any given time. To provide context to the measurements in the OFR, a bottom-up analysis can be carried out by applying laboratory SOA yields to the measured ambient SOA-forming gases that are entering the OFR.

The measured SOA formation after oxidation by $O_3$ and $NO_3$ is shown vs. the SOA predicted to form from measured precursor gases in Fig. 7. The measured SOA formation includes all ages greater than 0.7 eq. d for $O_3$-PAM and greater than 0.3 eq. d for $NO_3$-PAM, where most or all of the VOCs have reacted. For both oxidants, the data are scattered along the 1:1 line of equal measured and predicted SOA formation. This is in contrast to the analysis for OH oxidation in Palm et al. (2016), where a factor of 4.4 more SOA was formed from OH oxidation than could be explained by measured VOC precursors. As shown in that analysis, the additional SOA-forming gases in ambient air were likely S/IVOCs, where the SOA formation from S/IVOCs was 3.4 times larger than the source from VOCs. This conclusion was supported by unspeciated measurements of total S/IVOC concentrations (classified by volatility). SOA yields from S/IVOCs or any other sources are not required to explain SOA formation from $O_3$ or $NO_3$. This suggests that the majority of S/IVOCs in this ambient forest air generally did not contain C=C bonds, and therefore did not typically react with $O_3$ or $NO_3$ to produce SOA on atmospherically relevant time scales. This is consistent with expectations based on laboratory and ambient studies of MT and SQT oxidation products. Typical oxidation products include compounds such as pinic acid, pinonic acid, pinonaldehyde, caronaldehyde, and nopinone, none of which contain C=C double bonds (e.g., Calogirou et al., 1999b; Yu et al., 1999; Lee et al., 2006). As an example, the reaction rates of pinonaldehyde with OH, $O_3$, and $NO_3$ are $3.9 \times 10^{-11}$, $<2 \times 10^{-20}$, and $2.0 \times 10^{-14}$ $cm^3$ $molec^{-1}$ $s^{-1}$, respectively (Atkinson et al., 2006). These rates correspond to eq. lifetimes of 4.7 h, >579 d, and 29 d, respectively, showing that pinonaldehyde will typically only react with OH in the atmosphere or in the OFR under the conditions in this study.

While the measured and predicted SOA formation shown in Fig. 7 are consistent with each other, two main caveats limit the strength of the conclusions that can be drawn from this particular study. First, the amount and dynamic range of SOA formed from $O_3$ and $NO_3$ oxidation were relatively small, as were the total ambient

aerosol concentrations. This caused the SMPS+AMS measurement noise and variability to be larger relative to the total aerosol measurements than they would be for higher aerosol concentrations. Also, as only a small amount of new SOA was formed, the aerosol condensational sink remained relatively low for all measurements. According to the LVOC fate model, on average only 31% and 36% of LVOCs condensed to form SOA during $O_3$ and $NO_3$ oxidation, respectively (see Fig. S3). This required a correction of approximately a factor of 3 to correct measured SOA formation to what would occur in normal atmospheric conditions.

### 3.3    H:C and O:C ratios of SOA formed from oxidation of ambient air

Analysis of ambient high-resolution AMS spectra can be used to estimate the elemental composition of OA (Aiken et al., 2008; Canagaratna et al., 2015). When SOA is formed in the OFR, the OA that is sampled in the OFR output is a sum of preexisting ambient OA and any SOA produced from oxidation. At sufficiently high eq. ages, the sampled OA will also include the effects of heterogeneous oxidation. The amount of O, C, and H atoms added by oxidation can be calculated by subtracting the ambient elemental concentrations from those measured after aging. The amounts of each element added by oxidation can be used to determine the O:C and H:C elemental ratios of the SOA that is formed in the OFR.

The amounts of O and H vs. C added from OH oxidation are shown in Fig. 8. Slopes were fit to the data with positive net addition of C in order to determine the O:C and H:C of the SOA formed for the eq. photochemical age ranges of 0.1–0.4 (avg.=0.18) d, 0.4–1.5 (avg.=0.9) d, 1.5–5 (avg.=2.7) d, and 5–15 (avg.=10) d. The elemental O:C (H:C) ratios of the SOA mass formed in those ranges were 0.55 (1.60), 0.84 (1.44), 1.13 (1.36), and 1.55 (1.22). For data with ages of longer than several eq. days, O was added coincident with loss of C (i.e., negative x-intercept), which is likely due to heterogeneous oxidation leading to fragmentation/evaporation of preexisiting OA. This conclusion is reinforced by the evidence that for eq. OH ages greater than several days, heterogeneous oxidation resulted in a net loss of C when ambient MT concentrations were low (Fig. S10), but not for lower eq. ages. Similarly, George and Abbatt (2010) suggested that the lifetime of ambient OA with respect to heterogeneous OH oxidation is approximately two to three days. Therefore, the change in amounts of O, C, and H after several eq. days of oxidation will be a mix of heterogeneous change to preexisting OA and addition of new SOA. These effects of heterogeneous oxidation (i.e., x- and y-intercepts) are likely to be approximately the same for all data within each given age range, meaning the slopes fitted above are

independent of the heterogeneous processes and contain information about the elemental changes associated with the formation of varying amounts of SOA within each age range.

Analogous to Fig. 8, the amount of O and H vs. C added from $O_3$ and $NO_3$ oxidation are shown in Figs. 9–10. The SOA added from $O_3$ oxidation had O:C and H:C ratios of 0.50 and 1.61. The SOA added from $NO_3$ oxidation had O:C and H:C ratios of 0.39 and 1.60. This O:C value of 0.39 for $NO_3$ oxidation includes only the O atoms that were bound to the C backbone of the organic molecules, and excludes the two O atoms that are bound only to N in the $-ONO_2$ (nitrate) functional group (Farmer et al., 2010). If all O atoms in the nitrate functional group are included, the O:C of this added SOA mass was 0.44. Inclusion of only the carbon-bound oxygen of the nitrate functional group is more reflective of the carbon oxidation state, and is also what is typically reported for AMS O/C measurements (since the organic $-NO_2$ moeity is measured in the AMS as total nitrate and typically not separated from inorganic nitrate).

Heterogeneous oxidation was not expected to be a factor for the $O_3$ and $NO_3$ ages used in this work. This assumption was reinforced by the fact that no net loss of C was observed for these amounts of oxidation, even when ambient MT concentrations (and OA enhancement) were low, as shown in Figs. S11–12. This assumption is also consistent with previous research on lifetimes of OA components with respect to $O_3$ and $NO_3$ heterogeneous oxidation. For instance, several aldehydes were found to have a relatively long lifetime equivalent to approximately 2–8 days for $NO_3$ heterogeneous oxidation when calculated using 1 pptv ambient $NO_3$ (Iannone et al., 2011). Ng et al. (2016) summarized that reactive uptake of $NO_3$ into particles is slow for most molecules, with the exception of unsaturated or aromatic molecules, which were unlikely to be major components of the ambient OA in this remote forest (Chan et al., 2016). Although the lifetime of pure oleic acid (which contains a C=C bond) particles with respect to heterogeneous $O_3$ oxidation can be as short as tens of minutes (Morris et al., 2002), lifetimes for oleic acid in atmospheric particle organic matrices can be tens of hours to days (Rogge et al., 1991; Ziemann, 2005). Furthermore, the uptake coefficients for $O_3$ to react with saturated molecules are typically 1–2 orders of magnitude slower than for unsaturated molecules (de Gouw and Lovejoy, 1998). In summary, this previous research suggests that heterogeneous oxidation by $O_3$ or $NO_3$ may be important at higher eq. ages, but not for those achieved in the present work.

To put the O:C and H:C values of the SOA formed in the OFR in perspective, Van Krevelen diagrams of H:C vs O:C ratios for OA measured after OH, $O_3$, and $NO_3$ oxidation are shown compared to concurrent measurements of ambient OA in Fig. 11a–c, and summarized together in Fig. 11d. The effect of heterogeneous OH oxidation on preexisting aerosol is also shown as a line with a slope of -0.58. This line was fitted to the H:C vs. O:C of all OH-aged data where a net loss of C was observed (i.e., SOA formation was not observed and heterogeneous oxidation dominated). Generally speaking, less oxidized ("fresh") OA will lie in the upper left portion of a Van Krevelen plot, with higher H:C values and lower O:C values. Conversely, more oxidized ("aged") OA will move towards the lower right, with lower H:C values and higher O:C values (Heald et al., 2010; Ng et al., 2011). Shown in Fig. 11, the SOA formed from $O_3$, $NO_3$, and the lowest amount of OH aging (0.1–0.4 eq. days) was found at the upper left of the range occupied by ambient OA. As OH aging increased to higher ranges, the values of H:C decreased and the values of O:C increased, already moving beyond the local ambient range after 0.9 eq. days. At the higher ages, the H:C of the SOA formed lies at higher H:C values than those of the total OA measured after OH aging, which are closer to the trend of heterogeneous oxidation in the Van Krevelen space. This shows that SOA formed via gas-phase OH oxidation processes in an OFR has a higher H:C than the OA that results from heterogeneous oxidation, while both processes lead to similar increases in O:C. The net movement in the Van Krevelen space can be considered as starting at the ambient H:C and O:C and moving along two vectors: one vector along the heterogeneous oxidation line and another towards the H:C and O:C values of the new SOA formed in the gas phase, where the length of those two vectors are weighted by the amount of OA resulting from each process. When little SOA is formed, the H:C and O:C measured after oxidation lie along the heterogeneous oxidation line. When high amounts of SOA are formed, the H:C and O:C after oxidation shift to higher H:C values, lying closer to the curve defined by the H:C and O:C of SOA mass added in the OFR at the different age ranges (see Fig. S13).

While these two vectors describe the possible oxidation processes in the OFR, there may be other vectors (e.g., from condensed phase chemistry or reactive uptake) occurring in the atmosphere. As documented in Hu et al. (2016), SOA formation processes that require reactive uptake or within-particle non-radical chemistry (such as uptake of isoprene epoxydiols to form IEPOX-SOA) on time scales longer than the several minute residence time in the OFR are not captured with the OFR method used in this work. This is because the rate of reactive uptake and non-radical particle-phase chemistry do not speed up proportionally to increased OH and $HO_2$ (or $O_3$ or

NO$_3$). However, to our knowledge the only precursor for which reactive uptake of epoxides has been shown to be a major pathway is isoprene, which was a very minor precursor at this site (Karl et al., 2012). The formation of epoxides during MBO oxidation has been proposed to play at role during BEACHON-RoMBAS (Zhang et al., 2012). However, recent results suggest that formation of epoxides during MBO oxidation is not important in the atmosphere (Knap et al., 2016). Thus, at this time it is not clear whether any important SOA-forming processes in this environment are missed by the OFR setup, and this question should be investigated in future studies.

The H:C of the least oxidized SOA formed in the OFR from all oxidants was near 1.6. As discussed in Palm et al. (2016), SOA formation from OH oxidation in the OFR correlated with MT, and the S/IVOC sources of SOA may have been MT oxidation products or other related biogenic gases. Biogenic terpenes are composed of isoprene units, meaning they all have H:C of 1.6. Therefore, the SOA formed from the lowest eq. ages in the OFR was consistent with oxidation processes that add roughly 4–6 O atoms without removing net H atoms. Addition of –OH or –OOH functional groups after –H abstraction by OH radicals results in addition of O without loss of H, and are consistent with the RO$_2$+HO$_2$ reaction conditions that are expected during OH oxidation in the OFR (Kroll and Seinfeld, 2008; Ortega et al., 2016). OH can also add to a C=C bond, which could lead to addition of H atoms after oxidation. O$_3$ and NO$_3$ are expected to react with MT almost exclusively by addition to a C=C bond, which leads to addition of O without initial removal of H atoms (Atkinson and Arey, 2003). However, previous research has shown that many precursor gases, including aromatic molecules with initial H:C close to 1, can form SOA with H:C close to 1.6 (Chen et al., 2011; Chhabra et al., 2011; Canagaratna et al., 2015; Hildebrandt Ruiz et al., 2015). Therefore, H:C alone cannot provide direct evidence about the specific identities of precursor gases in ambient air. The SOA from O$_3$, NO$_3$, and 0.1–0.4 eq. days OH aging had H:C values similar to typical semi-volatile oxidized organic aerosol (SV-OOA), while the H:C of SOA from 0.4–1.5 eq. days or longer OH aging resembled low volatility oxidized organic aerosol (LV-OOA); these two types of SOA have been identified in ambient air at many locations (Jimenez et al., 2009; Canagaratna et al., 2015).

The relative time scales of oxidation and condensation in the OFR also need to be considered in order to properly interpret the H:C and O:C of the SOA mass formed in the OFR. In the atmosphere, once a molecule is oxidized to an LVOC that is able to condense onto a particle, lifetimes for condensation onto aerosols are on the order of several minutes (Farmer and Cohen, 2008; Knote et al., 2015; Nguyen et al., 2015). This is typically much shorter than the lifetimes for subsequent reaction with OH, O$_3$, or NO$_3$ of tens of minutes to several hours

or longer, so condensation will likely occur prior to further oxidation. In OFR oxidation experiments, the lifetime for subsequent oxidation of LVOCs is shortened proportional to the increase in oxidant concentration. However, the condensation lifetime does not scale with oxidant concentration, and remains roughly constant. At sufficiently high oxidant concentrations, LVOCs can be subjected to further oxidation steps that they would not

be subjected to in the atmosphere prior to having a chance to condense to form SOA. To compare SOA formation in the OFR vs. ambient air, these relative time scales are considered here as a function of both oxidant type and amount of oxidant exposure.

The lowest range of OH aging for which O:C and H:C values were measured was 0.1–0.4 (avg. 0.18) eq. d, which is 2.4–9.6 (avg. 4.3) eq. h of oxidation. Typical terpenes have lifetimes for reaction with OH on the order of tens

of minutes to several hours in the atmosphere (Atkinson and Arey, 2003), which is similar to this lowest eq. age range in the OFR. Typical terpene oxidation products have lifetimes ranging from 3.9 h (caronaldehyde; Alvarado et al., 1998) to 4.7 h (pinonaldehyde; Atkinson et al., 2006) to 11–13 h (nopinone; Atkinson and Aschmann, 1993; Calogirou et al., 1999a) to a computationally estimated 18–21 h (pinic and pinonic acid; Vereecken and Peeters, 2002). As a rough approximation, this suggests that the SOA formed in the OFR is likely

a result of approximately one or at most a few oxidation steps occurring to the molecules that enter the OFR (which may have already experienced one or more oxidation steps in the atmosphere prior to entering the OFR). The aging in this range strikes a balance between achieving enough oxidation to react all incoming precursors at least once while not reacting them an unrealistic number of times in the gas phase before allowing sufficient time for condensation. In the next age range of 0.4–1.5 (avg. 0.9) eq. d of OH aging, in which

the maximum OA enhancement occurred, some primary precursors are likely starting to be oxidized multiple times inside the OFR prior to condensation, while some oxidation products will still be oxidized only ~1–2 times. The SOA formed in this range may represent SOA formed from multiple generations of chemistry. At higher ages in the OFR, the aerosol is likely mainly modified by heterogeneous oxidation, with a small contribution from condensation of highly oxidized products. This OA at the highest ages resembles ambient OA found in

remote locations (Jimenez et al., 2009; Chen et al., 2015). Indeed, OFRs have previously been used to study heterogeneous oxidation processes (George et al., 2008; Smith et al., 2009).

For $O_3$ and $NO_3$ oxidation, the oxidants will react only with C=C double-bond-containing gases. The major MT and SQT species at this field site all contain only a single C=C bond (isoprene and minor MT and SQT species

contain two). Subsequent reaction lifetimes of oxidation products with these oxidants will likely be longer than the lifetime for condensation onto particles. For example, the lifetimes for pinonaldehyde with respect to $O_3$ and $NO_3$ oxidation are >579 d, and 29 d, respectively (Atkinson et al., 2006). Therefore, we can approximate that multiple generations of oxidation are not dominant for SOA formation when investigating $O_3$ or $NO_3$ oxidation in the OFR at this site. This is consistent with previous chamber SOA formation experiments that suggested that first-generation oxidation products dominate SOA formation from $O_3$ oxidation of a variety of biogenic compounds with a single C=C bond, rather than products of later generations of oxidation (Ng et al., 2006). The SOA formed via $O_3$ or $NO_3$ oxidation in the OFR is likely formed from reaction with primary VOCs and a small subset of their reaction products that still contain C=C bonds, such as the α-pinene oxidation product campholenic aldehyde (Kahnt et al., 2014). This SOA should be representative of typical atmospheric SOA formation processes.

### 3.4    Particulate organic nitrate (pRONO$_2$) formation from NO$_3$ oxidation of ambient air

In addition to estimating the elemental composition of OA, the AMS can also be used to estimate the amount of inorganic vs. organic nitrate in submicron aerosols (Farmer et al., 2010; Fry et al., 2013). The ratio of $NO_2^+$ to $NO^+$ fragment ions produced by thermal decomposition on the AMS vaporizer and electron impact ionization depends on the type of nitrate. $NH_4NO_3$ typically produces a ratio of approximately 0.3-1, while particulate organic nitrate (pRONO$_2$), in which the $-ONO_2$ functional group is covalently bonded to the carbon backbone (R) through an oxygen atom, typically produces a ratio ~2-3 times lower (Fry et al., 2009; Bruns et al., 2010; Farmer et al., 2010; Liu et al., 2012; Day et al., 2017). The measured $NO_2^+$ to $NO^+$ ratio is a linear combination of these two chemical components. Using this principle, the $NO_3$ measured by the AMS was split into the estimated fractions of $NH_4NO_3$ and pRONO$_2$ according to the method described in Fry et al. (2013). For the instrument in this work, ratios of 0.3 and 0.13 were used for the $NO_2^+$ to $NO^+$ ratios of $NH_4NO_3$ and pRONO$_2$, respectively (Fry et al., 2013).

A two-night example of both ambient and NO$_3$-radical aged aerosol on Aug. 20–22 is shown in Fig. 12. In ambient air, the majority of NO$_3$ aerosol was organic. After oxidation in the OFR, different behavior was seen on the two nights shown. On the first night, mainly inorganic nitrate was produced, as evidenced by the higher $NO_2^+$ to $NO^+$ ratio, the formation of $NH_4$ aerosol, and the relatively small amount of SOA formed. On the second

night, pRONO$_2$ was produced, as evidenced by the lower NO$_2^+$/NO$^+$ ratio, a lack of NH$_4$ aerosol formation, and substantial SOA formation. The organic nitrate formation and SOA formation also roughly tracked the ambient MT concentrations.

These two distinct behaviors in the NO$_3$-OFR were likely controlled by ambient RH. There was a competition between thermal dissociation of injected N$_2$O$_5$ to produce NO$_3$+NO$_2$ (favored at high temperatures and low RH) and the hydrolysis of N$_2$O$_5$ on wetted OFR walls to produce HNO$_3$ (favored at low temperatures and high RH). When hydrolysis occurred rapidly, then there was a sharp decrease in N$_2$O$_5$ concentrations. The NO$_3$ radical concentrations were also greatly reduced, and thus fewer NO$_3$ radicals were available to react with ambient gases (e.g., MT) to produce pRONO2. HNO$_3$ reacted with NH$_3$ in ambient air or evaporating from OFR surfaces to produce NH$_4$NO$_3$. The results shown in Fig. 12 illustrate this behavior, with NO$_3$ radical exposure being reduced while NH$_4$NO$_3$ was produced during the first night. Despite the presence of similar MT concentrations on both nights, little SOA was produced on the first night. Future applications could include heating of the OFR slightly above ambient temperatures in order to prevent hydrolysis of N$_2$O$_5$ on the OFR walls. Inhibiting NH$_4$NO$_3$ formation artifacts would be especially critical for data interpretation if measuring aerosol enhancements with only non-chemical instruments such as an SMPS.

Despite this complex chemistry, information about the chemical composition of pRONO$_2$ formed from real atmospheric precursors can still be derived from times when conditions favored pRONO$_2$ formation. Shown in Fig. 13 is the mass of organic –ONO$_2$ added vs. SOA added from oxidation by each of the three oxidants. Substantial formation of pRONO$_2$ was observed only for NO$_3$ radical oxidation, and not for O$_3$ or OH oxidation. This was expected, since ambient NO$_x$ concentrations were generally low (0.5–4 ppb; Ortega et al., 2014), and the NO$_3$ oxidation experiment was the only one with an added source of reactive nitrogen. The slopes of Fig. 13 represent the ratio of –ONO$_2$ to the rest of the organic molecules in pRONO$_2$. In this study, the slope after NO$_3$ radical oxidation was 0.10, which is similar to the range of 0.1–0.18 found in previous chamber studies of NO$_3$ oxidation of terpenes (Fry et al., 2009, 2011; Boyd et al., 2015). To put this in context, if every SOA molecule formed in the OFR contained a single –ONO$_2$ group (with its mass of 62 g mol$^{-1}$), then the molecular mass of the full pRONO$_2$ molecules would be an average of 620 g mol$^{-1}$ (giving the slope of 62 g mol$^{-1}$ / 620 g mol$^{-1}$ = 0.10 in Fig. 13). Alternatively, if all molecules are assumed to have a mass of 200 or 300 g mol$^{-1}$, then 32% or 48% of the molecules, respectively, would contain a –ONO$_2$ functional group (assuming no molecules contain more than

one –ONO$_2$ group). Again, this result is roughly consistent with previous research. For the fraction of OA composed of pRONO$_2$ in NO$_3$+$\beta$-pinene SOA, Fry et al. (2009) estimated 32–41% (assuming an average molecular weight of 215–231 g mol$^{-1}$), Fry et al. (2014) estimated 56% (assuming 214 g mol$^{-1}$), and Boyd et al. (2015) estimated 45–68% (assuming 200-300 g mol$^{-1}$).

**4      Conclusions**

In situ SOA formation from ambient pine forest air after oxidation by OH, O$_3$, or NO$_3$ radicals was measured using an OFR for the first time. SOA formation from these real ambient mixes of aerosol and SOA precursors was measured semi-continuously, capturing diurnal and daily changes in the relative ambient concentrations of SOA precursor gases. In general, more SOA was formed from the precursors present in nighttime air than in

daytime air for all three oxidants. At all times of day, OH oxidation produced approximately 4 times more SOA than O$_3$ or NO$_3$ oxidation. The O:C and H:C ratios of the SOA formed by O$_3$, NO$_3$, and several eq. hours of OH oxidation was similar to the oxidation levels of ambient OA.

The OFR is a tool that can be used to measure the total SOA formation potential of ambient air at any given time, and how that potential changes with time, whether or not the SOA precursor gases are measured and/or

speciated. As discussed in Palm et al. (2016), ambient VOC concentrations alone could not explain the amount of SOA formed in the OFR by OH oxidation. Instead, SOA was likely being formed from S/IVOCs that entered the OFR. In contrast, the quantity of measured VOCs was sufficient to explain the amount of SOA formed from O$_3$ and NO$_3$ oxidation; closure between measured and predicted SOA formation in an OFR was achieved. In other words, O$_3$ and NO$_3$ oxidation of the ambient S/IVOCs do not appear to produce appreciable amounts of SOA.

This suggests that the ambient S/IVOCs tend not to have double bonds.

While this work does not investigate the source of the S/IVOCs, one possibility is that they are oxidation products of primary VOCs (e.g., MT or SQT). The primary VOCs could be emitted upwind of the site, and by the time the molecule enters the OFR, the double bond(s) will have reacted, leaving an oxidation product that reacts further with OH but not O$_3$ or NO$_3$. If the lifetime for further reaction of these oxidation products is

slower than the lifetime for the double-bond-containing primary emissions, then the oxidation products will build up in the atmosphere. Under this hypothesis, such S/IVOC compounds are not new or unexpected sources

of SOA. In most regional and global models, they would already be implicitly accounted for, by tracking the emissions of the primary VOCs which have corresponding overall SOA yields. In this work, we consider only the primary VOCs that are measured to be entering the OFR, not the integrated sum of upwind emissions that were emitted into the air that eventually entering the OFR after some degree of ambient photochemical processing.

SOA formation in the OFR takes a snapshot of the atmosphere, which consists of a mix of primary emissions and their oxidation products at various stages of oxidative progress. For this study, those snapshots demonstrate that for OH oxidation, only approximately a quarter of the SOA-forming gases are in the form of primary VOC, while for $O_3$ and $NO_3$ oxidation almost all are in the form of primary VOC. It also suggests that for these precursor mixtures, multi-generational chemistry plays a major role in the overall amount of SOA formed from

OH oxidation (and much less so for $O_3$ and $NO_3$).

If these SOA-forming S/IVOCs do not react with ambient $O_3$ or $NO_3$, they will build up in the atmosphere during the night when OH is absent. When the sun rises and OH is produced, a sudden burst of SOA formation might be expected. However, this coincides with dilution of gases and particles due to convective vertical mixing, potentially offsetting such new SOA formation and making it difficult to observe it without detailed chemical

and boundary layer dynamics measurements and/or modeling. These OFR measurements and analysis elucidate the presence and properties of S/IVOCs in the atmosphere, and highlight the need for more measurements and modeling of such gases in order to better understand ambient SOA formation. This work also demonstrates the utility of the OFR as a tool for studying SOA formation from all three major atmospheric oxidants.

**Acknowledgements**

We thank William H. Brune for discussions and contributions including the initial development of the PAM OFR. We thank US NSF grants AGS-1243354 and AGS-1360834, NOAA grant NA13OAR4310063, U.S. DOE ASR Program (Office of Science, BER, DE-SC0011105 & DE-SC0016559), Austrian Science Fund (FWF) project number L518-N20, and the US EPA (STAR 83587701-0) for partial support for this research. BBP acknowledges support from a CIRES Graduate Student Research Fellowship and a US EPA STAR Graduate Fellowship (FP-91761701-0).

This work has not been formally reviewed by the US EPA. The views expressed are solely those of the authors, and the US EPA does not endorse any products or commercial services mentioned in this work. LK acknowledges support from DOC-fFORTE-fellowship of the Austrian Academy of Science. The National Center

for Atmospheric Research (NCAR) is sponsored by the National Science Foundation. AMO acknowledges a fellowship from the DOE SCGP Fellowship Program (ORAU, ORISE). KJZ was supported by NASA grant NNX09AE12G. We are grateful to Alex Guenther and Jim Smith of NCAR for co-organizing the BEACHON-RoMBAS field campaign, and to the USFS Manitou Experimental Forest Observatory for site support.

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

**Figures**

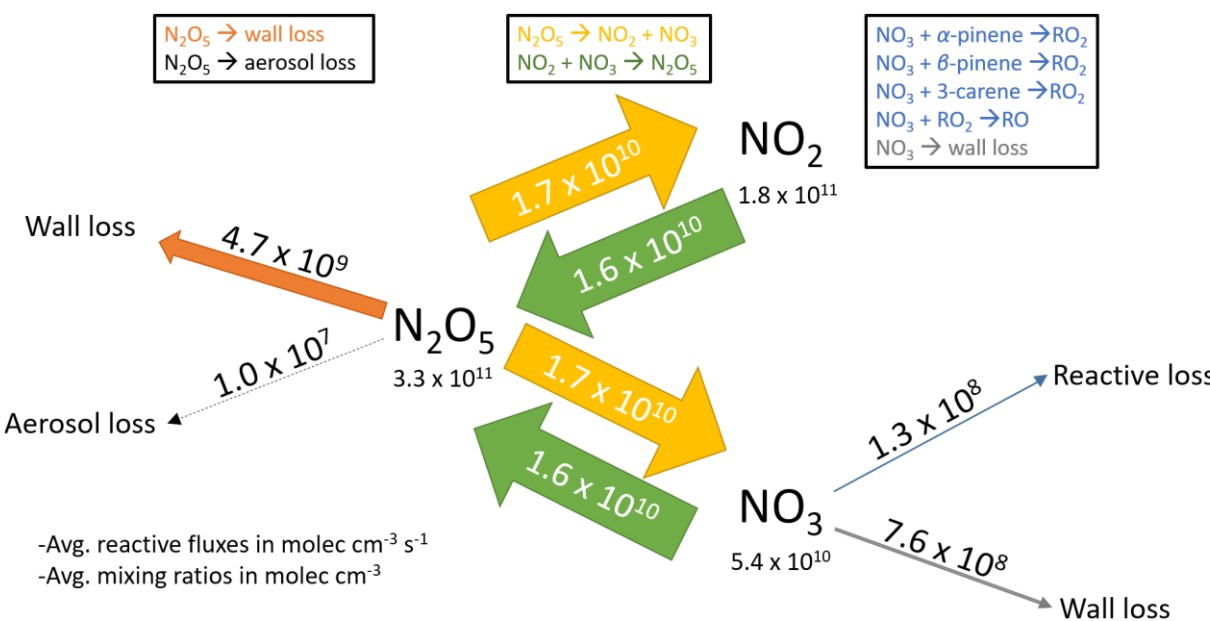

**Fig. 1.** Typical average mixing ratios and reactive fluxes for the major reactions in the NO3-OFR when injecting $N_2O_5$ to investigate SOA formation from $NO_3$ oxidation. These reactive fluxes resulted from running the model with inputs of 25°C, 50% RH, 50 ppb $O_3$, 2 ppb $NO_2$, 1.5 ppb $NO_3$, 50 ppb $N_2O_5$, 0.75 ppb total MT, and a rate constant for $N_2O_5$ uptake to aerosol surfaces of $3\times10^{-5}$ s$^{-1}$. Reaction arrow widths are sized relative to their average reactive fluxes. Reactions that were included in the model (shown in Table S1) but with smaller average rates are not shown here.

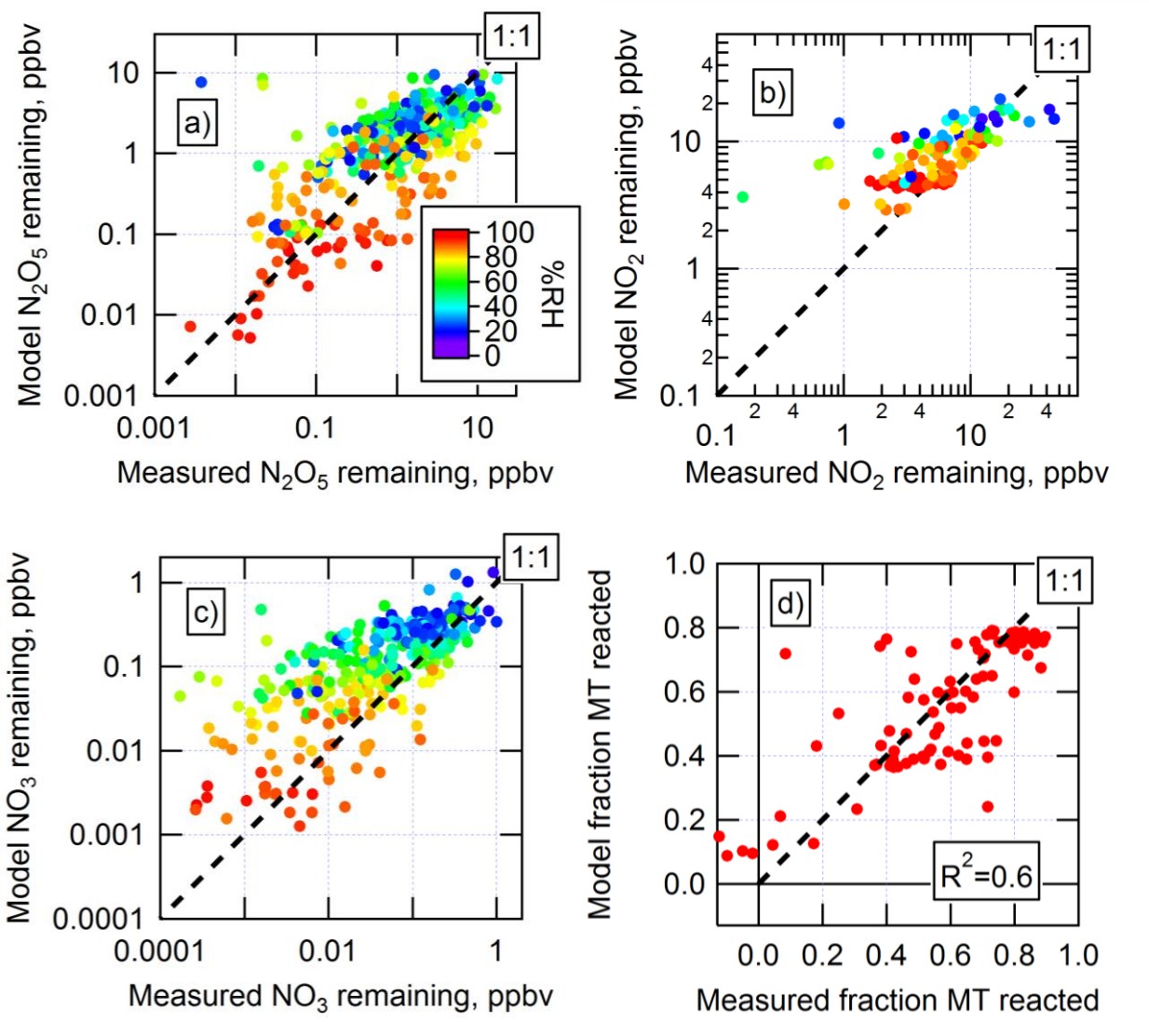

**Fig. 2.** Modeled vs. measured a) N₂O₅, b) NO₂, c) NO₃, and d) fraction of ambient MT reacted with NO₃ in the output of the NO₃-OFR.

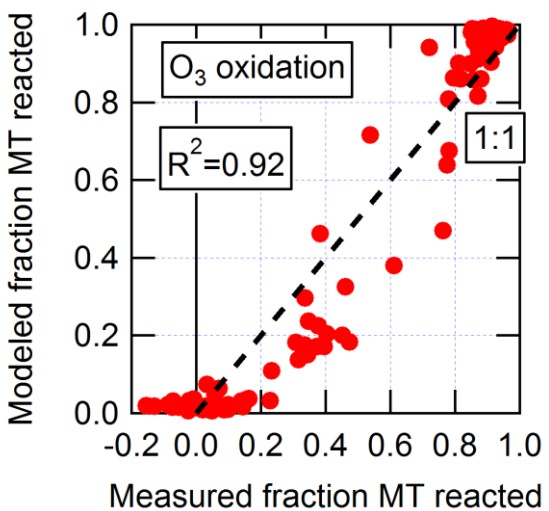

**Fig. 3.** Modeled vs. measured fraction MT reacted by $O_3$ oxidation in the $O_3$-OFR.

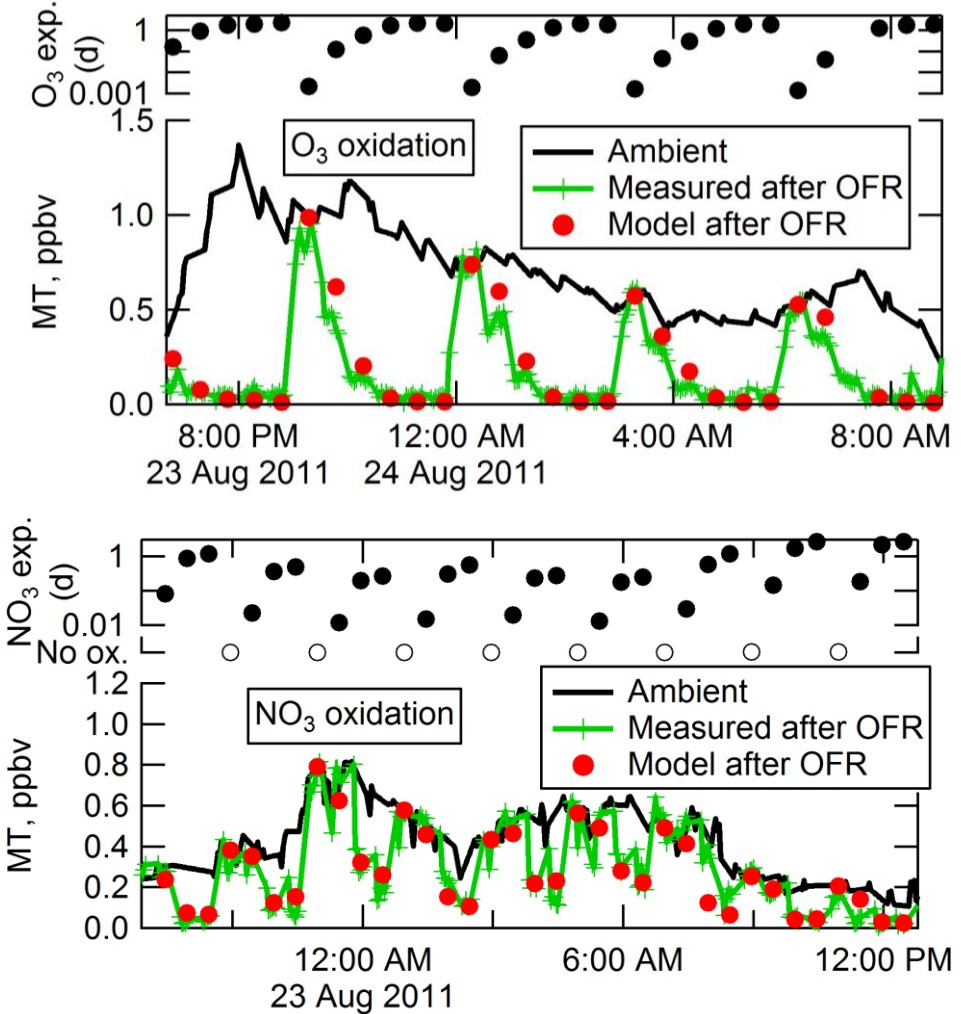

**Fig. 4.** Ambient, measured remaining, and modeled remaining MT from $O_3$ oxidation on Aug. 23–24 (top) and $NO_3$ oxidation on Aug. 22–23 (bottom) in the OFR. Modeled $O_3$ and $NO_3$ exposures are also shown. The amount of oxidation was cycled from no added oxidant (no MT reacted) to maximum oxidation (most or all MT reacted) in repeated 2–3 h cycles.

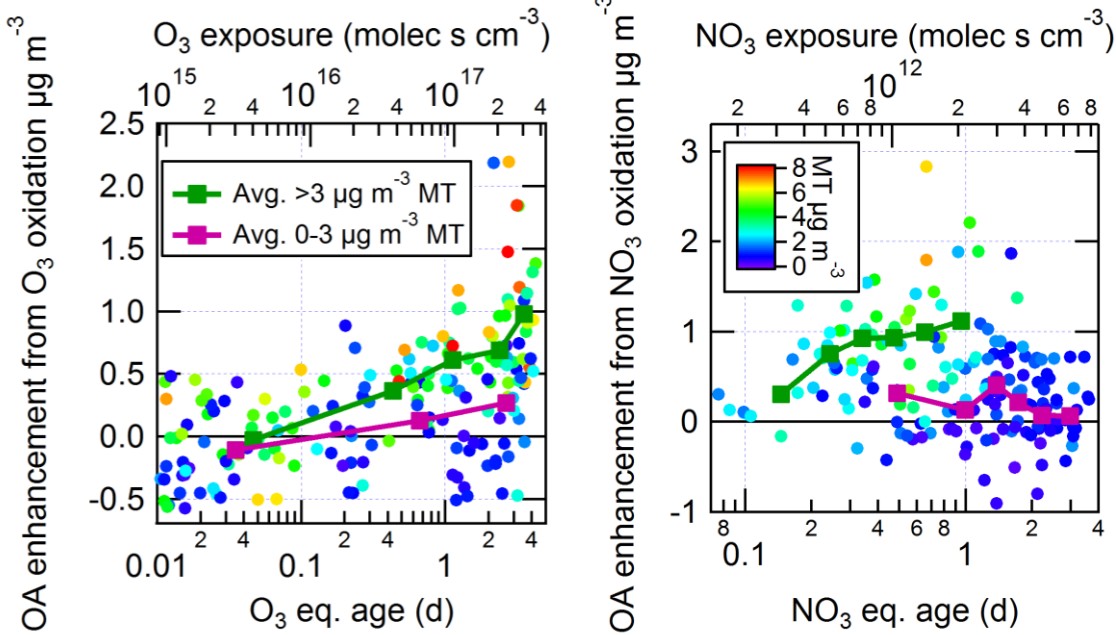

**Fig. 5.** OA enhancement from oxidation of ambient air by $O_3$ (left) and $NO_3$ (right) as a function of oxidant exposure. Data are colored by ambient in-canopy MT concentrations and include the LVOC fate correction. Binned averages for times when ambient MT concentrations were either below or above 3 µg m$^{-3}$ (0.66 ppb) are also shown, illustrating the positive relationship between OA enhancement and MT concentrations at the higher oxidant concentrations.

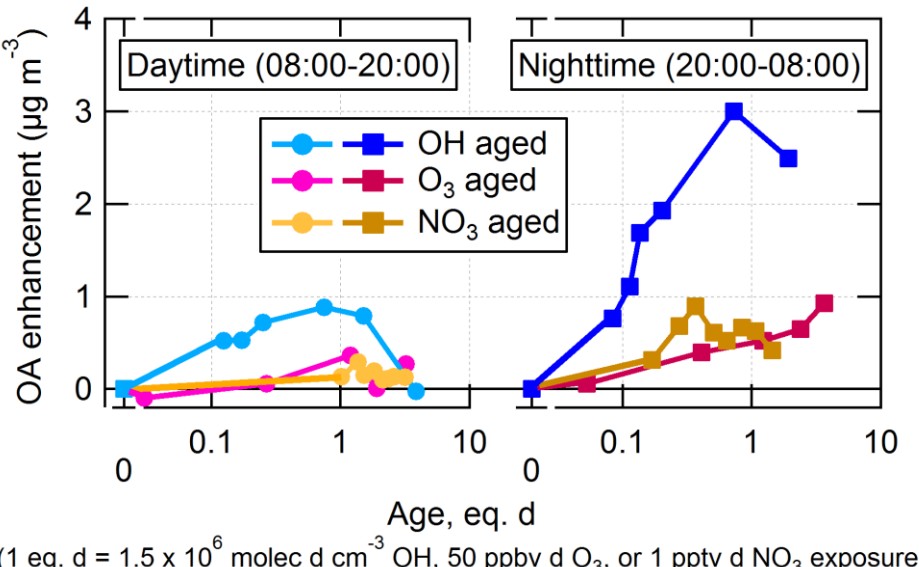

**Fig. 6.** OA enhancement vs. age in eq. d for OH, $O_3$, and $NO_3$ oxidation, separated into daytime (08:00–20:00 LT) and nighttime (20:00–08:00 LT) data. All data is LVOC fate corrected. OH oxidation produced several-fold more OA enhancement than $O_3$ and $NO_3$ oxidation. OH-aged OA enhancement data is taken from Palm et al. (2016), and shows data only for <5 eq. d aging where the LVOC fate correction could be applied.

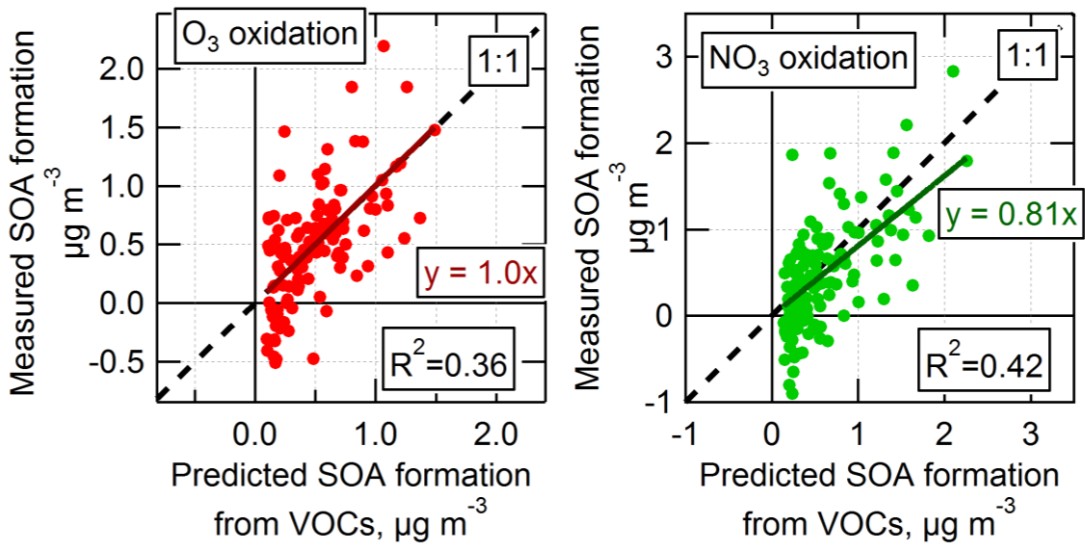

**Fig. 7.** Measured vs. predicted SOA formation for $O_3$ and $NO_3$ oxidation in an OFR. The measured SOA formation includes the LVOC fate correction, and includes all ages greater than 0.7 eq. d for $O_3$-PAM and greater than 0.3 eq. d for $NO_3$-PAM. Predicted SOA formation was estimated by applying published chamber SOA yields to the mass of VOCs predicted by the model to be oxidized in the OFR (see Sect. 2.3 for details).

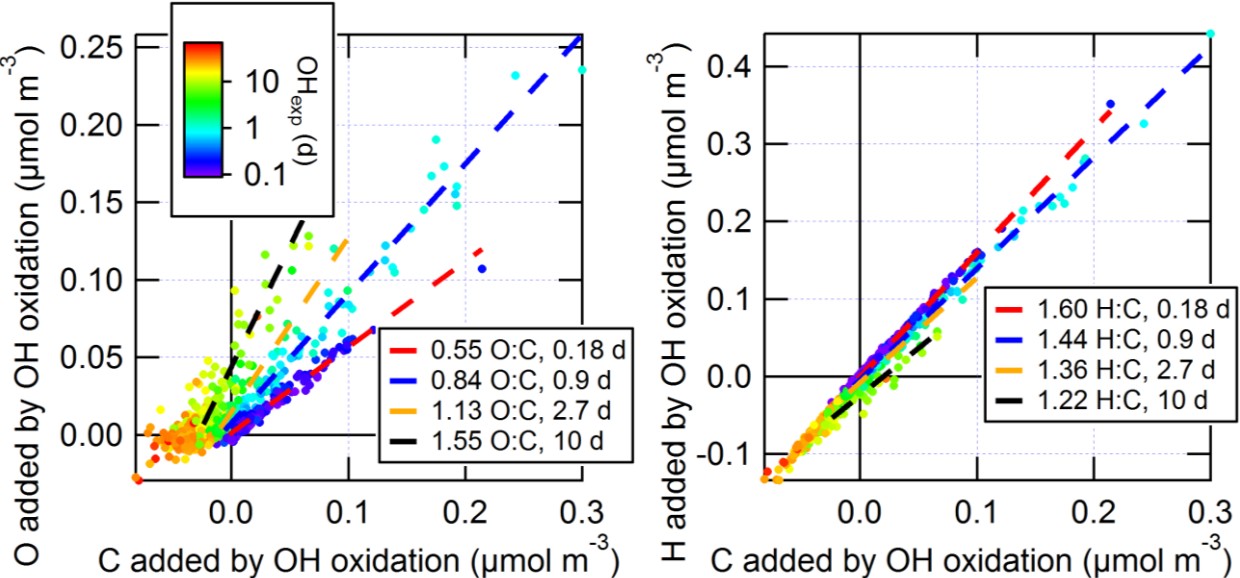

**Fig. 8.** Scatter plots of μmol m$^{-3}$ O and μmol m$^{-3}$ H added per μmol m$^{-3}$ C added from OH oxidation of ambient air in the OFR. Slopes are fit to the photochemical age ranges of 0.1–0.4 (avg.=0.18) d, 0.4–1.5 (avg.=0.9) d, 1.5–5 (avg.=2.7) d, and 5–15 (avg.=10) d, showing that the atomic O:C(H:C) ratios of the SOA mass formed in those ranges were 0.55 (1.60), 0.84 (1.44), 1.13 (1.36), and 1.55 (1.22), respectively. At higher ages, heterogeneous oxidation led to loss of C and H and little to no loss of O.

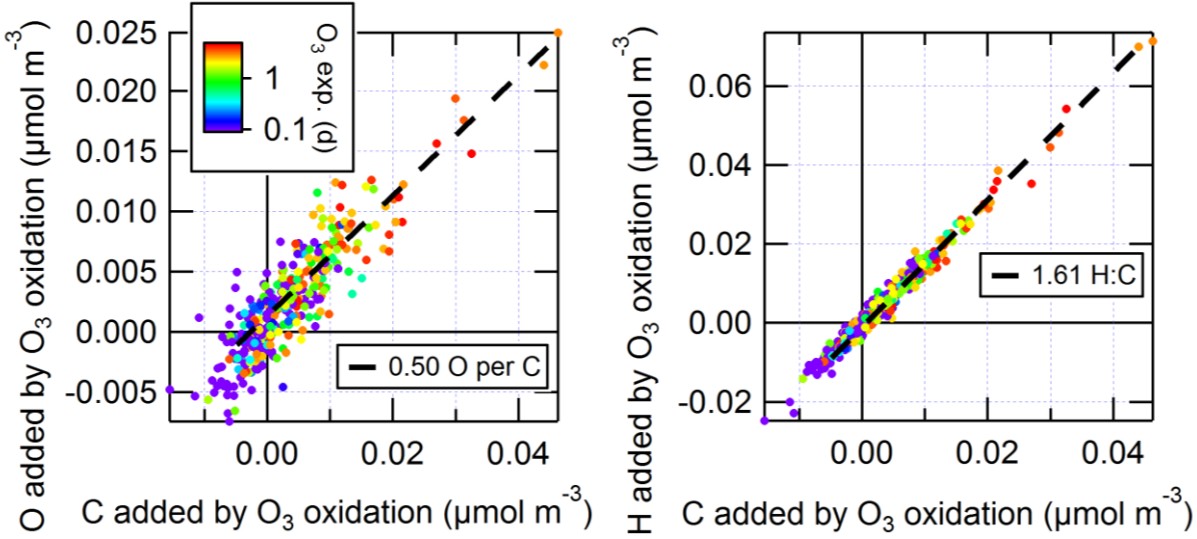

**Fig. 9.** Scatter plots of μmol m$^{-3}$ O and μmol m$^{-3}$ H added per μmol m$^{-3}$ C added from O$_3$ oxidation of ambient air in the OFR. Data are colored by eq. d of O$_3$ exposure. The slopes show that the atomic O:C (H:C) ratio of the SOA mass formed was 0.50 (1.61). The slopes did not change with increasing photochemical age.

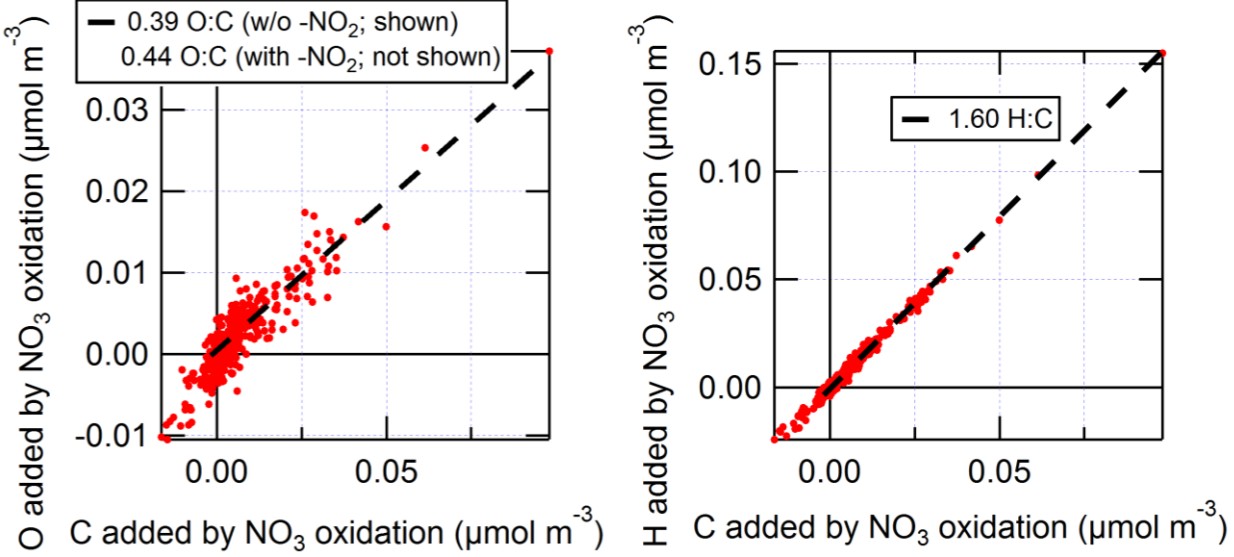

**Fig. 10.** Scatter plots of μmol m$^{-3}$ O and μmol m$^{-3}$ H added per μmol m$^{-3}$ C added from NO$_3$ oxidation of ambient air in the OFR. The amount of O added is shown without including the O from the –NO$_2$ group, since those O atoms do not affect the oxidation state of C. The slopes show that the atomic O:C(H:C) ratio of the SOA mass formed was 0.39 (1.60). The slopes did not change with increasing NO$_3$ exposure. Contrary to Figs. 8–9, data are not colored by NO$_3$ exposure. The ranges of NO$_3$ exposure achieved during daytime vs. nighttime were unequal (Figs. 5-6, S12), obscuring any trend of OA enhancement vs. eq. age.

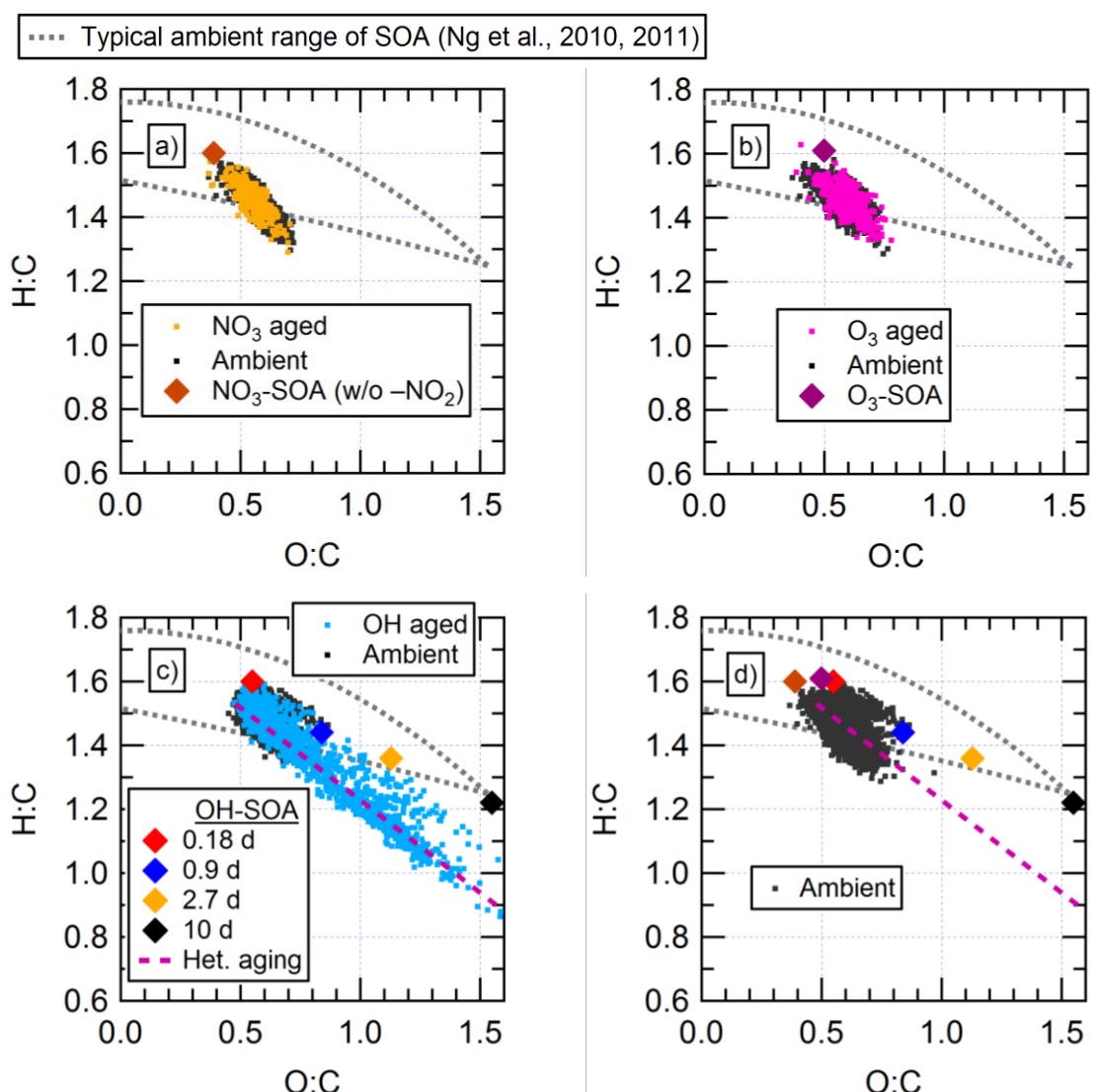

**Fig. 11.** Van Krevelen diagrams of H:C vs. O:C ratios of OA after oxidation by a) $NO_3$, b) $O_3$, and c) OH along with concurrent ambient ratios. The H:C and O:C ratios of the new SOA mass formed in the OFR (i.e., the slopes from Figs. 8–10) are shown for each oxidant (diamonds), and are summarized in d) compared with all ambient measurements. For data where no net C addition was observed after OH oxidation, the slope along which heterogeneous OH oxidation transforms the ambient OA is shown (purple dashed line).

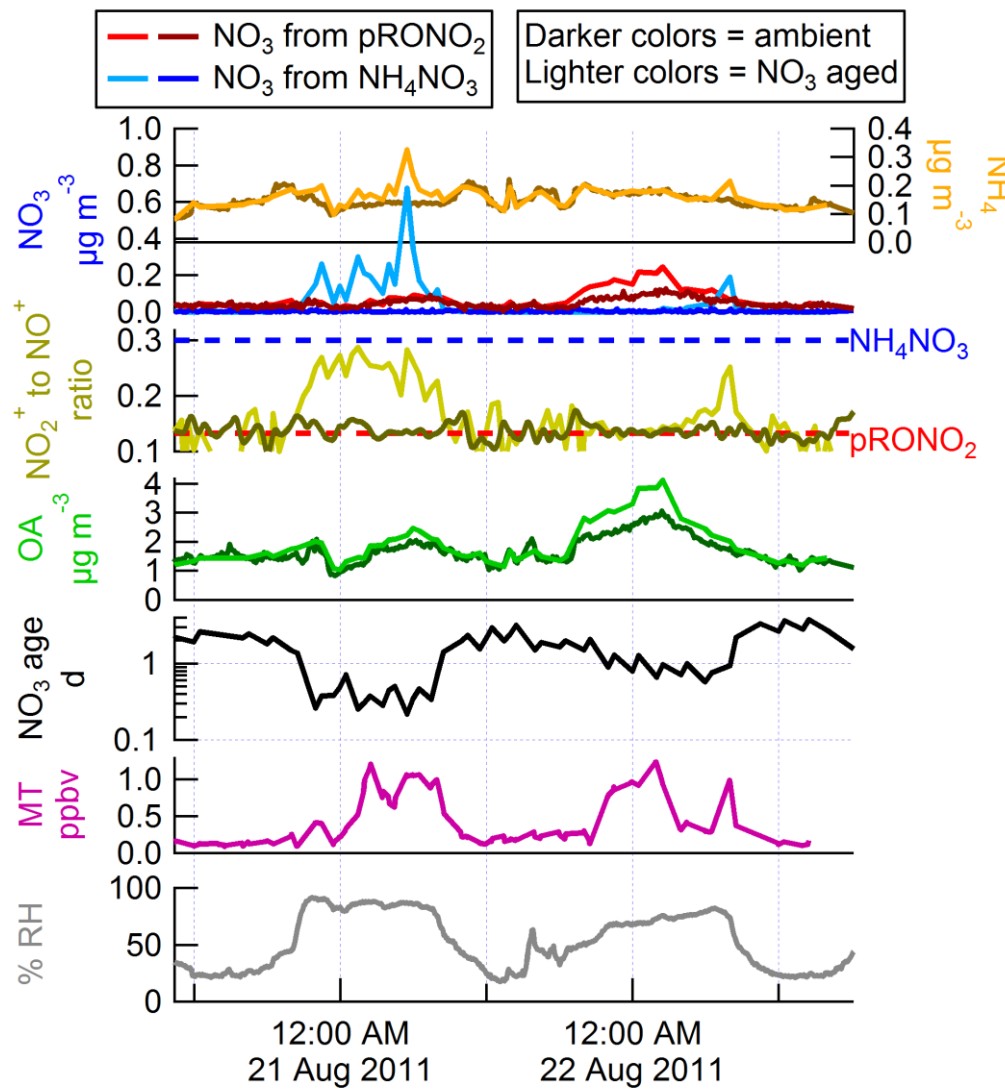

**Fig. 12.** Example time series of OA, NH₄, and NO₃ (split into pRONO₂ and NH₄NO₃) aerosol measurements after NO₃ oxidation in the OFR, compared to ambient aerosol, NO₂⁺ to NO⁺ ratio, model-derived eq. age of NO₃ oxidation, MT concentration, and RH measurements. Production of both NH₄NO₃ and pRONO₂ was observed at different times, which appears to depend on changes in experimental conditions.

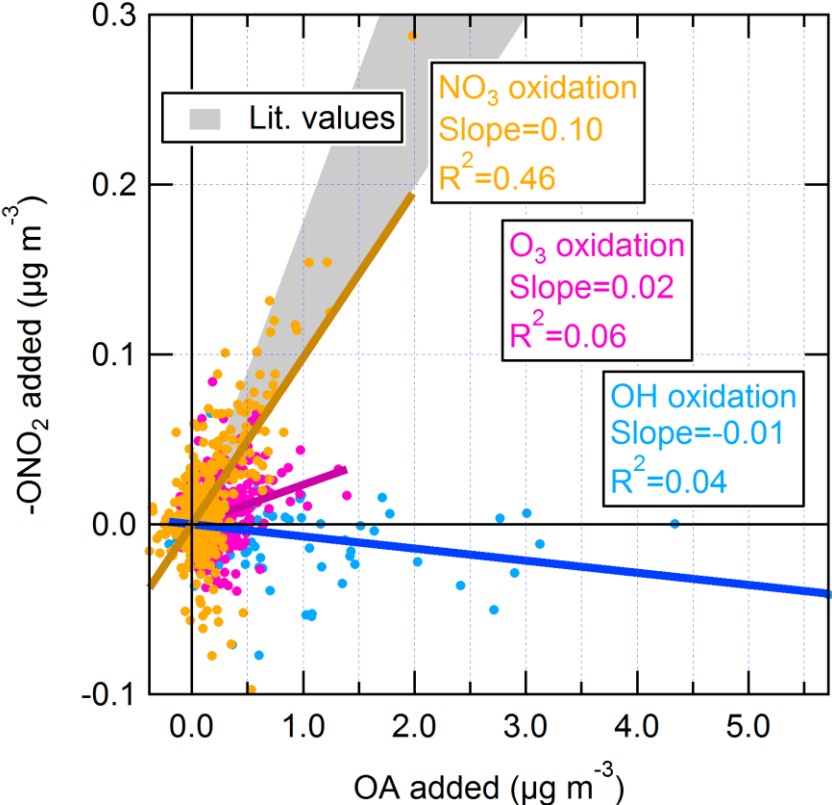

**Fig. 13.** Organic –ONO$_2$ mass added vs. OA added from OH, O$_3$, and NO$_3$ oxidation in an OFR. No pRONO$_2$ formation was observed (or expected) from OH or O$_3$ oxidation under the experimental conditions. The slope of 0.10 from NO$_3$ oxidation is consistent with previous chamber measurements (shown in grey), which range from approximately 0.1–0.18 (Fry et al., 2009, 2011; Boyd et al., 2015).