# Peer review of "Secondary organic aerosol formation from in situ OH, O$_3$, and NO$_3$ oxidation of ambient forest air in an oxidation flow reactor"

_Atmospheric Chemistry and Physics, 2016_

## Referee Comment (RC1) · Anonymous Referee #1 · 16 Mar 2017

Summary and Overall Recommendation:

This well-written and impressive manuscript summarizes oxidation flow reactor (OFR) experiments aimed at studying in situ SOA formation from ambient pine forest air during the BEACHON-ROMBAS campaign after oxidation by OH, O3, and NO3 radicals. Since SOA formation was measured semi-continuously during this study, the authors were able to capture diurnal and daily changes. More SOA was formed from precursors present in nighttime air than in the daytime air for all 3 oxidations. Interestingly, OH oxidation produced ∼ 4 times more SOA than NO3 and O3 oxidation at all times of day. O:C and H:C ratios of the SOA formed by O3, NO3 and several eq. hours of OH oxidation yielded similar oxidation levels of ambient organic aerosol (OA). The

authors previously demonstrated that ambient VOC concentrations alone could not explain the amount of SOA formed in the OFR by OH oxidation. This behavior was likely due to SOA being formed from semivolatile/intermediate volatility organic compounds (S/IVOCs) that entered the OFR. However, for SOA formed from O3 and NO3 oxidation, the measured VOCs were found in the present study to be sufficient in explaining the amount of SOA formed in the OFR. More specifically, this means that for O3 and NO3 oxidation of ambient S/IVOCs does not yield appreciable SOA amounts. The difference between the OH and O3/NO3 OFR experiments provides some support for their hypothesis that ambient S/IVOCs generally lacking double bonds in their structures (especially since double bonds in VOCs emitted upwind of the site are likely already oxidized before they enter the OFR). Using ambient mixtures in this study provides important insights into SOA formation potential and chemical evolution in the real atmosphere, and thus, this work will be of high interest to the larger atmospheric chemistry community. I only have a few minor comments below that I kindly ask the authors to address before publication. As a result, I recommend that this manuscript be accepted with minor revisions noted below.

1.) My biggest comment is related to timescales in the OFR for multi-phase chemical processes. Since the authors appear to justify that their OFR experiments can produce similar oxidation states (O:C ratios) in OA found in the atmosphere, my question is this a result of the "correct" processes that actually occur in the atmosphere? Besides for heterogeneous oxidation (through OH oxidation), what about aqueous-phase processes such as accretion or decomposition reactions of epoxides and or hydroperoxides? There is a lot of work published now by the Caltech, UNC, Oberlin College, and other groups that have shown epoxides are really important in aqueous-phase chemical processes. Recently, the Harvard (Martin) and UNC groups have shown that multi-phase chemical reactions of hydroperoxides could be important as well (Liu et al., 2016, PCCP; Riva et al., 2017, Atmos. Environ.). There is evidence from this site that even MBO oxidation products can undergo aqueous-phase reactions within aerosol to yield organosulfates (Zhang et al., 2012, ES&T). I'm not sure authors can

really address this issue now, but I think some discussion needs to be included that acknowledges that these processes may explain some part of ambient oxidation states, which can't be reflected on the reaction timescales of the OFR.

2.) In section 2.2 of the experimental methods section, can the authors provide more information or clarify on how the ambient might or might not change upon entering the OFR? Specifically, is it drier in the OFR compared to the ambient RH? If the RHs aren't the same, how might this affect the interpretation of the results?

3.) I'm curious if the authors know how hydroperoxides behave in their OFR? Do they photolyze quite easily due to the UV radiation you are using? How might this affect the interpretation of the results?

―――――――――――――――――――――――

---

## Referee Comment (RC2) · Anonymous Referee #2 · 21 Mar 2017

This work describes the first field observations of in-situ OH, O3, and NO3 exposures to ambient air using an oxidative flow reactor. This is highly important work in the field of atmospheric chemistry today, with extensive field and lab studies being performed to better understand the chemical mechanisms and potential to form (or fragment) secondary organic aerosol. Observations here are conducted in a forested environment with biogenic precursor gases (monoterpene dominant) and highlight the dominance of OH oxidation chemistry, but show potential for O3 and NO3 reactions with C=C bond VOC species at night. Several studies have been performed using a similar method since the 2011 BEACON-RoMBAS study described here, making the analysis and results of this study very relevant for upcoming manuscripts for this research team and

others. In-situ NO3 chemistry and modeling is especially novel.

Specific comments to be addressed:

Pg. 4, Line 22: Discussing MT's here, but haven't defined how these are measured, if cumulative MT's by PTR, or summed by GC/MS.

Pg. 5, Line 5: Please provide average concentration increases for "moderate increases" of NOx, CO, and anthro VOCs. Also, what anthro VOCs?

Pg. 5, Line 20: Are periods with very high local winds excluded from the analysis?

Pg. 6: The thorough explanation of NO3 exposure estimates here and in supplemental material is appreciated. It seems worth considering how representative one equivalent day of NO3 aging would be of atmospheric conditions. Given the typical diel pattern of NO3, and relatively low concentrations, would it ever be expected that a whole day's worth of oxidation could occur prior to further oxidation from OH?

Pg.8, Line 22: Can further argument be provided for the assumption in this modeled correction (of no fragmentation for O3 or NO3 reaction LVOC products)? I'm wondering to what extent does the assumption drive conclusions? Figure 5 suggests lower OA concentrations at 2-3 days NO3 eq. aging compared to 1 day eq. aging.

Pg. 9, line 1: The acronym for sesquiterpene (SQT) has not yet been defined.

Pg. 10, line 15: The negative values in Figure 2d for the fraction of monoterpenes reacted, along with the instances of OFR output MT concentrations that exceed ambient levels shown in Figure S7, should be mentioned. Can this be attributed to instrument uncertainty, or are there other factors at play that give these apparent MT generation events?

Pg. 10, line 24: Change "didn't" to "did not".

Pg. 12, Line 6: please provide average daytime MT+SQT concentration and average nighttime MT+SQT concentration here.

Pg. 12, line 19: In Figure 6, there is an uptick in OA enhancement with the highest level of O3 oxidation for the nighttime air. However, in Figure S7 it appears that the MTs are largely depleted prior to reaching this extent of aging. Would this suggest that something beyond the measured monoterpenes is contributing to SOA formation from O3 oxidation at these highest levels of aging?

Pg. 13, Line 11: abstract says factor of 3.4. Here is states factor of 4.4. Are these numbers referring the same discrepancy?

Pg. 18, line 24: Change "formed from primary VOCs" to "formed from reaction with primary VOCs".

Pg. 20, line 13: Please explain further where 620 g mol-1 is coming from.

Figure 5: This method of binning seems to limit comparison of low and high monoterpene conditions at the same levels of oxidation. Particularly for NO3, why are there not average values for the high monoterpene case at high levels of NO3 eq. age?

Supplemental Information

Fig S3: Should reiterate in figure caption that these fractional fates are modeled, not measured. Additionally, it seems that the fraction of LVOCs condensing on the aerosol will decrease slightly at higher NO3 exposures. Would this be due to a greater frequency of fragmentation reactions occurring as opposed to functionalization?

Fig S6: why higher NO3 exposures on the limited data points on Aug9-10?

Fig S8: Which quantile averages are being shown by the black trace?

Table S2: revisit for formatting

---

## Author Comment (AC1) · 1 Apr 2017

**Response to reviewers for "Secondary organic aerosol formation from in situ OH, O₃, and NO₃**
**oxidation of ambient pine forest air in an oxidation flow reactor."**

**B. B. Palm, J. L. Jimenez, et al.**

We thank the reviewers for their comments on our paper. To facilitate the review process we have
copied the reviewer comments in black text. Our responses are in regular blue font. We have responded
to all the referee comments and made alterations to our paper (**in bold text**).

**Anonymous Referee #1**

Overview

R1.0. This well-written and impressive manuscript summarizes oxidation flow reactor (OFR) experiments
aimed at studying in situ SOA formation from ambient pine forest air during the BEACHON-ROMBAS
campaign after oxidation by OH, O₃, and NO₃ radicals. Since SOA formation was measured semi-
continuously during this study, the authors were able to capture diurnal and daily changes. More SOA
was formed from precursors present in nighttime air than in the daytime air for all 3 oxidations.
Interestingly, OH oxidation produced ~ 4 times more SOA than NO₃ and O₃ oxidation at all times of day.
O:C and H:C ratios of the SOA formed by O₃, NO₃ and several eq. hours of OH oxidation yielded similar
oxidation levels of ambient organic aerosol (OA). The authors previously demonstrated that ambient
VOC concentrations alone could not explain the amount of SOA formed in the OFR by OH oxidation. This
behavior was likely due to SOA being formed from semivolatile/intermediate volatility organic
compounds (S/IVOCs) that entered the OFR. However, for SOA formed from O₃ and NO₃ oxidation, the
measured VOCs were found in the present study to be sufficient in explaining the amount of SOA
formed in the OFR. More specifically, this means that for O₃ and NO₃ oxidation of ambient S/IVOCs does
not yield appreciable SOA amounts. The difference between the OH and O₃/NO₃ OFR experiments
provides some support for their hypothesis that ambient S/IVOCs generally lacking double bonds in their
structures (especially since double bonds in VOCs emitted upwind of the site are likely already oxidized
before they enter the OFR). Using ambient mixtures in this study provides important insights into SOA
formation potential and chemical evolution in the real atmosphere, and thus, this work will be of high
interest to the larger atmospheric chemistry community. I only have a few minor comments below that I
kindly ask the authors to address before publication. As a result, I recommend that this manuscript be
accepted with minor revisions noted below.

R1.1. My biggest comment is related to timescales in the OFR for multi-phase chemical processes. Since
the authors appear to justify that their OFR experiments can produce similar oxidation states (O:C
ratios) in OA found in the atmosphere, my question is this a result of the "correct" processes that
actually occur in the atmosphere? Besides for heterogeneous oxidation (through OH oxidation), what
about aqueous-phase processes such as accretion or decomposition reactions of epoxides and or
hydroperoxides? There is a lot of work published now by the Caltech, UNC, Oberlin College, and other
groups that have shown epoxides are really important in aqueous-phase chemical processes. Recently,
the Harvard (Martin) and UNC groups have shown that multi-phase chemical reactions of
hydroperoxides could be important as well (Liu et al., 2016, PCCP; Riva et al., 2017, Atmos. Environ.).
There is evidence from this site that even MBO oxidation products can undergo aqueous-phase
reactions within aerosol to yield organosulfates (Zhang et al., 2012, ES&T). I'm not sure authors can
really address this issue now, but I think some discussion needs to be included that acknowledges that these processes may explain some part of ambient oxidation states, which can't be reflected on the
reaction timescales of the OFR.

We thank the reviewer for pointing out this caveat. We have included the following text as a new
paragraph starting after page 16, line 22:

**"While these two vectors describe the possible oxidation processes in the OFR, there may be other**
**vectors (e.g., from condensed phase chemistry or reactive uptake) occurring in the atmosphere. As**
**documented in Hu et al. (2016), SOA formation processes that require reactive uptake or within-**
**particle non-radical chemistry (such as uptake of isoprene epoxydiols to form IEPOX-SOA) on time**
**scales longer than the several minute residence time in the OFR are not captured with the OFR**
**method used in this work. This is because the rate of reactive uptake and non-radical particle-phase**
**chemistry do not speed up proportionally to increased OH and $HO_2$ (or $O_3$ or $NO_3$). However, to our**
**knowledge the only precursor for which reactive uptake of epoxides has been shown to be a major**
**pathway is isoprene, which was a very minor precursor at this site (Karl et al., 2012). The formation of**
**epoxides during MBO oxidation has been proposed to play at role during BEACHON-RoMBAS (Zhang**
**et al., 2012). However, recent results suggest that formation of epoxides during MBO oxidation is not**
**important in the atmosphere (Knap et al., 2016). Thus, at this time it is not clear whether any**
**important SOA-forming processes in this environment are missed by the OFR setup, and this question**
**should be investigated in future studies."**

R1.2. In section 2.2 of the experimental methods section, can the authors provide more information or
clarify on how the ambient might or might not change upon entering the OFR? Specifically, is it drier in
the OFR compared to the ambient RH? If the RHs aren't the same, how might this affect the
interpretation of the results?

To address this comment, we have added the following text to the experimental methods section at
page 5, line 16:

**"The OFR was located on top of the measurement trailer in order to sample ambient air directly**
**without using an inlet. Therefore the temperature and RH inside the OFR were the same as ambient**
**conditions, with the exception of minor heating from the UV lamps mounted inside the OH-OFR (up to**
**~2°C heating at the highest lamp settings, and ~0.5$^{o}$C at the settings producing the most SOA; Li et al.,**
**2015). No heating occurred during $O_3$ or $NO_3$ modes. Thus RH within the OFR was the same or slightly**
**lower than ambient, depending on the operating mode."**

R1.3. I'm curious if the authors know how hydroperoxides behave in their OFR? Do they photolyze quite
easily due to the UV radiation you are using? How might this affect the interpretation of the results?

Non-OH chemistry, such as photolysis of hydroperoxides, has indeed been investigated via modeling in
Peng et al. (2016). That investigation concluded that for a wide variety of gases and for OH-OFR
conditions in BEACHON-RoMBAS, reactions with OH dominated over other possible reactions, including
$O(^1D)$, $O(^3P)$, $O_3$, and photolysis at 185 nm or 254 nm. This was also the case for OH oxidation at other
field campaigns where ambient air was oxidized in the OFR. Peng et al. (2016) illustrated that non-OH
reactions can become significant under certain circumstances, such as very low RH, high external OH
reactivity, or when the gases involved are particularly reactive towards a non-OH pathway. However,
these conditions are more commonly found in laboratory studies, where they can also be avoided by carefully designing such experiments. Peng et al. (2016) also investigated photolysis of SOA, and found
that while photolysis could affect a small but non-negligible percentage of SOA, photolysis of SOA across
the lifetime of particles in the atmosphere would play a much larger role.

To address this comment, we have moved the sentence "The gas-phase $HO_x/O_x$ chemistry inside the OFR
has also been investigated with kinetic modeling (Li et al., 2015; Peng et al., 2015, 2016)." from page 5,
lines 12-13, to page 5, line 25, and altered it to read:

**"The gas-phase $HO_x/O_x$ chemistry and possible non-OH chemistry inside the OFR was investigated**
**with kinetic modeling (Li et al., 2015; Peng et al., 2015, 2016). For the wide variety of compounds**
**investigated in Peng et al. (2016), reactions with OH dominated over other possible reactions,**
**including $O(^1D)$, $O(^3P)$, $O_3$, and photolysis at 185 nm or 254 nm, under the conditions of OH oxidation**
**in the OFR during this campaign."**

**Anonymous Referee #2**

Overview

R2.0. This work describes the first field observations of in-situ OH, O3, and NO3 exposures to ambient air
using an oxidative flow reactor. This is highly important work in the field of atmospheric chemistry
today, with extensive field and lab studies being performed to better understand the chemical
mechanisms and potential to form (or fragment) secondary organic aerosol. Observations here are
conducted in a forested environment with biogenic precursor gases (monoterpene dominant) and
highlight the dominance of OH oxidation chemistry, but show potential for O3 and NO3 reactions with
C=C bond VOC species at night. Several studies have been performed using a similar method since the
2011 BEACON-RoMBAS study described here, making the analysis and results of this study very relevant
for upcoming manuscripts for this research team and others. In-situ NO3 chemistry and modeling is
especially novel. Specific comments to be addressed:

R2.1. Pg. 4, Line 22: Discussing MT's here, but haven't defined how these are measured, if cumulative
MT's by PTR, or summed by GC/MS.

We have changed the sentence starting at page 4, line 21, to: "VOC concentrations at the site
**(quantified using proton-transfer-reaction time-of-flight mass spectrometry; PTR-TOF-MS)** varied on a
diurnal cycle…"

R2.2. Pg. 5, Line 5: Please provide average concentration increases for "moderate increases" of NOx, CO,
and anthro VOCs. Also, what anthro VOCs?

We have changed the text at page 5, line 5 to: **"…leading to moderate increases in NO$_x$ (up to ~5 ppbv**
**from ~2 ppbv), CO (up to ~140 ppbv from ~100 ppbv), and anthropogenic VOCs (e.g., benzene up to**
**~50 pptv from ~20 pptv, and toluene up to ~150 pptv from ~50 pptv) during the late afternoon and**
**evening (Fry et al., 2013; Ortega et al., 2014)."**

R2.3. Pg. 5, Line 20: Are periods with very high local winds excluded from the analysis?

We have added the following text to the manuscript at page 5, line 20:

**"The data were not screened for high local wind speeds. However, periods of high wind speeds were**
**infrequent during the campaign, and the influence of local winds was likely tempered by the fact that**
**the OFR was located within the canopy of the forest."**

R2.4. Pg. 6: The thorough explanation of NO3 exposure estimates here and in supplemental material is
appreciated. It seems worth considering how representative one equivalent day of NO3 aging would be
of atmospheric conditions. Given the typical diel pattern of NO3, and relatively low concentrations,
would it ever be expected that a whole day's worth of oxidation could occur prior to further oxidation
from OH?

The reviewer touches on a very important point, which is that NO$_3$ concentrations in the atmosphere are
much more variable than those of OH or O$_3$. This means that the eq. NO$_3$ ages calculated assuming an
average of 1 pptv of NO$_3$ in this work need to be interpreted in the context of that assumption, which is
only strictly applicable to this research site. Other sites may have much more or less average ambient
NO$_3$. We had already made this point in the paragraph starting on page 6, line 24. To more strongly make the point that, even for a given location, the $NO_3$ concentrations can be variable from one night to
the next, we have altered the text starting on page 7, line 3 to read:

**"Estimated eq. $NO_3$ ages from this study are therefore shown simply for a common metric of**
**comparison for all of the data during this study, interpretable in terms of the average chemistry**
**occurring at the BEACHON site. Interpretation of measurements at other sites would need to be**
**adjusted to local $NO_3$ concentrations."**

R2.5. Pg.8, Line 22: Can further argument be provided for the assumption in this modeled correction (of
no fragmentation for O3 or NO3 reaction LVOC products)? I'm wondering to what extent does the
assumption drive conclusions? Figure 5 suggests lower OA concentrations at 2-3 days NO3 eq. aging
compared to 1 day eq. aging.

To address this comment, we have added the following text to page 8, line 24:

**"This assumption is reinforced by the fact that for the highest $O_3$ and $NO_3$ eq. ages achieved in this**
**work, no net decrease of OA was observed when SOA-forming gases were not present (see Sect. 3.2.1**
**and Fig. 5). If fragmentation reactions in the gas phase (or from heterogeneous oxidation) were**
**important for the range of eq. ages studied here, observations would show a net loss of OA at the**
**highest eq. ages when SOA-forming gases (e.g., MT) were not present."**

Regarding the lack of SOA formation observed at the highest $NO_3$ ages in Fig. 5, those data points were
coincident with low ambient MT concentrations (all blue on the MT concentration color bar), so little to
no SOA formation was expected. This comment is also addressed by our response to R2.14 below.

R2.6. Pg. 9, line 1: The acronym for sesquiterpene (SQT) has not yet been defined.

We thank the reviewer for catching this mistake. We have changed the page 9, line 1 instance of SQT to
**"sesquiterpenes (SQT)"**.

R2.7. Pg. 10, line 15: The negative values in Figure 2d for the fraction of monoterpenes reacted, along
with the instances of OFR output MT concentrations that exceed ambient levels shown in Figure S7,
should be mentioned. Can this be attributed to instrument uncertainty, or are there other factors at play
that give these apparent MT generation events?

To address this comment, we have added the following text at page 10, line 15:

**"The scatter in the measurements is thought to be due mainly to incomplete and/or variable mixing of**
**the injected $N_2O_5$ flow into the sampled ambient air (see Sect. S1 for more details), with some**
**contribution from measurement variability at low ambient MT concentrations."**

We have also added the following text to the end of the Fig. S7 caption:

**"Note that the ambient MT were sampled through a separate inlet within the canopy, several meters**
**from the OFR. Short periods of higher MT concentrations measured through the OFR (at low $O_3$**
**exposures) may be due to spatial heterogeneity in ambient MT concentrations within the canopy."**

R2.8. Pg. 10, line 24: Change "didn't" to "did not".

Done.

R2.9. Pg. 12, Line 6: please provide average daytime MT+SQT concentration and average nighttime
MT+SQT concentration here.

We have changed the text at page 12, line 6 to:

**"This is consistent with the general increase in MT and SQT (average of 1.1 and 0.04 ppbv in the**
**canopy during nighttime, and 0.4 and 0.03 ppbv during daytime, respectively) and related precursor**
**concentrations in the shallower nighttime boundary layer."**

We have also clarified the related text in Sect. 2.1. The VOC concentrations quoted in the original text
referred to the measurements at 25 m, above the forest canopy. As shown in Palm et al. (2016), the in-
canopy concentrations were higher, and those are the concentrations that are relevant to this analysis.
Therefore, the text on page 4, line 24 was changed to:

**"During BEACHON-RoMBAS, the concentration of MBO+isoprene in the forest canopy ranged from**
**about 2 ppb during daytime to 0.4 ppb at nighttime (see Palm et al., 2016)."**

The text at page 5, line 2 was changed to:

**"MT concentrations in the canopy spanned from 0.4 ppb during the day to 1.1 ppb at night, on**
**average."**

R2.10. Pg. 12, line 19: In Figure 6, there is an uptick in OA enhancement with the highest level of O3
oxidation for the nighttime air. However, in Figure S7 it appears that the MTs are largely depleted prior
to reaching this extent of aging. Would this suggest that something beyond the measured
monoterpenes is contributing to SOA formation from O3 oxidation at these highest levels of aging?

The apparent uptick in OA enhancement at the highest $O_3$ eq. ages is most likely a result of
measurement variability due to the limited number of measurements in each eq. age bin. To address the
possibility that the $O_3$ ages used in this study were not high enough to lead to SOA formation from non-
VOC precursors, we have changed the text at page 12, line 19 to read:

**"Such molecules would typically not react appreciably with $O_3$ or $NO_3$ over the range of eq. ages**
**achieved in this work, but will still react with OH and may lead to SOA formation. Future $O_3$ and $NO_3$**
**oxidation studies should include higher eq. age ranges in order to investigate if additional SOA could**
**be formed from ambient precursors at higher ages."**

R2.11. Pg. 13, Line 11: abstract says factor of 3.4. Here is states factor of 4.4. Are these numbers
referring the same discrepancy?

We have clarified the relationship between these two numbers by changing the text at page 13, line 10
to:

**"This is in contrast to the analysis for OH oxidation in Palm et al. (2016), where a factor of 4.4 times**
**more SOA was formed from OH oxidation than could be explained by measured VOC precursors. As**
**shown in that analysis, the additional SOA-forming gases in ambient air were likely S/IVOCs, where**
**the SOA formation from S/IVOCs was 3.4 times larger than the source from VOCs. This conclusion was**
**supported by unspeciated measurements of total S/IVOC concentrations (classified by volatility)."**

R2.12. Pg. 18, line 24: Change "formed from primary VOCs" to "formed from reaction with primary
VOCs".

We have changed this text to read:

**"formed from reaction with primary VOCs."**

R2.13. Pg. 20, line 13: Please explain further where 620 g mol-1 is coming from.

We have clarified this point by changing the text at page 20, line 11 to:

**"To put this in context, if every SOA molecule formed in the OFR contained a single $-ONO_2$ group (with**
**its mass of 62 g mol$^{-1}$), then the molecular mass of the full $pRONO_2$ molecules would be an average of**
**620 g mol$^{-1}$ (giving the slope of 62 g mol$^{-1}$ / 620 g mol$^{-1}$ = 0.10 in Fig. 13)."**

R2.14. Figure 5: This method of binning seems to limit comparison of low and high monoterpene
conditions at the same levels of oxidation. Particularly for NO3, why are there not average values for the
high monoterpene case at high levels of NO3 eq. age?

The range of eq. $NO_3$ ages achieved in the OFR was strongly influenced by ambient temperature, which
controlled the equilibrium between $N_2O_5$ and $NO_2+NO_3$ from the injected $N_2O_5$. During nighttime (when
MT concentrations were higher) it was colder and less $NO_3$ exposure was realized in the OFR. During
daytime (with lower MT concentrations), warm ambient temperatures led to more $NO_3$ exposure. To
make this clearer, we have added the following text on page 11, line 21:

**"As seen in Fig. 5 (and in Fig. 6 below), lower eq. $NO_3$ ages were achieved when MT concentrations**
**were higher, and higher eq. $NO_3$ ages were achieved when MT concentrations were lower. This was**
**because the higher MT concentrations occurred during nighttime, when lower ambient temperatures**
**shifted the equilibrium towards $N_2O_5$ and away from $NO_2+NO_3$ (from the injected $N_2O_5$), meaning**
**lower $NO_3$ exposures were realized in the OFR."**

Due to the data limitations, we did not bin data by multiple MT concentrations for day or night;
however, the non-binned data points are shown as well (and colored by MT) in order to give a sense of
the relationship between SOA formation and MT concentrations for similar oxidation levels. That
relationship is also borne out in the measured vs. modeled discussion in Section 3.2.2 and Fig. 7.

Supplemental Information:

R2.15. Fig S3: Should reiterate in figure caption that these fractional fates are modeled, not measured.
Additionally, it seems that the fraction of LVOCs condensing on the aerosol will decrease slightly at
higher NO3 exposures. Would this be due to a greater frequency of fragmentation reactions occurring as
opposed to functionalization?

We have changed the first line of the Fig. S3 caption from "Fractional fates" to **"Modeled fractional**
**fates"** as suggested. The slightly lower apparent fraction that condenses on particles at higher eq. $NO_3$
ages is a result of the slightly lower condensational sink (i.e., lower aerosol concentrations) during the
daytime when those high eq. ages were achieved (see also response to comment R2.14). Fragmentation
at high exposures was not included in the model, as described in Sect. 2.3 and in response to comment
R2.5.

R2.16. Fig S6: why higher NO3 exposures on the limited data points on Aug9-10?

We have changed the last sentence of the Fig. S6 caption to read:

**"For these examples, the amount of injected $N_2O_5$ was held roughly constant (with a higher constant**
**value injected on Aug. 9–10)."**

R2.17. Fig S8: Which quantile averages are being shown by the black trace?

We have changed the last sentence of the Fig. S8 caption to read:

**"Quantile averages of OA enhancement per ppbv MT are shown for each oxidant, with error bars**
**corresponding to the standard error of the mean of each quantile."**

R2.18. Table S2: revisit for formatting.

We thank the reviewer for pointing out the issue with the formatting of line numbers. It has been fixed.

Other Changes:

1: On page 7, line 20, we changed the typo "Scanning Particle Mobility Analyzer" to **"Scanning Mobility**
**Particle Sizer"**.

2: The author list in the Supplemental Information was changed to match the author list in the main
paper.

3: We have corrected Fig. S1 to reflect a small change in the FLUENT model results. The new figure is
presented here:

[Figure]

**Fig. S1. Normalized residence time distributions in the OFR as a function of normalized residence time**
**(1 = avg. residence time of each distribution). The FLUENT model was used to calculate residence**
**times for 1 nm particles (with Brownian motion) and 100 nm particles (without Brownian motion) for**
**the OFR configuration without the inlet plate to represent conditions used during BEACHON-RoMBAS.**
**These distributions are compared to the bis(2-ethylhexyl) sebacate (BES) particle residence time**
**distribution measured with the inlet plate installed in Lambe et al. (2011) and to the ideal plug flow**
**distribution (where all particles have equal residence time calculated as the OFR volume divided by**

[revised manuscript text omitted]